# Auxiliary Cross-Modal Common Representation Learning with Triplet Loss Functions for Online Handwriting Recognition

## Abstract

Common representation learning (CRL) learns a shared embedding between two or more modalities to improve in a given task over using only one of the modalities. CRL from different data types such as images and time-series data (e.g., audio or text data) requires a deep metric learning loss that minimizes the distance between the modality embeddings. In this paper, we propose to use the triplet loss, which uses positive and negative identities to create sample pairs with different labels, for CRL between image and time-series modalities. By adapting the triplet loss for CRL, higher accuracy in the main (time-series classification) task can be achieved by exploiting additional information of the auxiliary (image classification) task. Our experiments on synthetic data and handwriting recognition data from sensor-enhanced pens show an improved classification accuracy, faster convergence, and a better generalizability.

## 1 Introduction

Cross-modal retrieval such as common representation learning (CRL) Peng et al. (2017) for learning across two or more modalities (i.e., image, audio, text and 3D data) has attracted a lot of attention recently. It can be applied in a wide range of applications such as multimedia management Lee et al. (2020) and identification Sarafianos et al. (2019). Extracting information from several modalities and adapting the domain with cross-modal learning allows to use information in all domains Ranjan et al. (2015). CRL, however, remains challenging due to the *heterogeneity gap* (inconsistent representation forms of different modalities) Huang et al. (2020).

A limitation of CRL is that most approaches require the availability of all modalities at inference time. However, in many applications certain data sources are only available during training by means of elaborate laboratory setups Lim et al. (2019). For instance, consider a human pose estimation task that uses inertial sensors together with RGB videos during training. A camera setup might not be available at inference time due to bad lighting conditions or other application-specific restrictions. This requires a model that allows inference on the main modality only, while auxiliary modalities may only be used to improve the training process (as they are not available at inference time) Hafner et al. (2022). *Learning using privileged information* Vapnik & Izmailov (2015) is one approach in the literature that describes and tackles this problem. During training it is assumed that in addition to $X$ additional information $X^*$, the *privileged information*, is available which is, however, not present in the inference stage Momeni & Tatwawadi (2018).

For CRL, we need a deep metric learning (DML) technique that aims to transform training samples into feature embeddings that are close for samples that belong to the same class and far apart for samples from different classes Wei et al. (2016). As DML requires no model update (simply fine-tuning for training samples of new classes), DML is an interesting approach for continual learning Do et al. (2019). Typical DML methods use simple distances (e.g., Euclidean distance) but also highly complex distances (e.g., canonical correlation analysis Ranjan et al. (2015) and maximum mean discrepancy Long et al. (2015)). While CRL learns representations from all modalities, single-modal learning commonly uses pair-wise learning. The triplet loss Schroff et al. (2015) selects a positive and negative triplet pair for a corresponding anchor and forces the positive pair distance to be smaller than the negative pair distance.

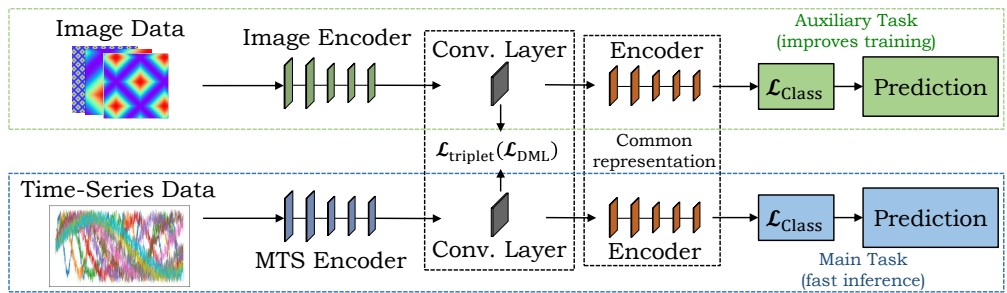

Figure 1: **Method overview:** Common representation learning between image and time-series data using the triplet loss based on DML functions to improve the time-series classification task.

While research of triplet selection for single-modal classification is very advanced Do et al. (2019), pair-wise selection for CRL is investigated for specific applications only Zhen et al. (2015); Lee et al. (2020); Zhang & Zheng (2020).

One exemplary application for cross-modal learning is handwriting recognition (HWR). HWR can be categorized into offline and online HWR. Offline HWR such as optical character recognition (OCR) deals with the analysis of the visual representation of handwriting only, but cannot be applied for real-time recognition applications Fahmy (2010). Online HWR works on different types of spatio-temporal signals and can make use of temporal information such as writing speed and direction Plamondon & Srihari (2000). Many recording systems make use of a stylus pen together with a touch screen surface Alimoglu & Alpaydin (1997). Systems for writing on paper are only prototypical Chen et al. (2021); Schrapel et al. (2018); Wang et al. (2013); Deselaers et al. (2015) and cannot be applied for real-world applications. A novel sensor-enhanced pen based on inertial measurement units (IMUs) enables new applications for writing on normal paper. This pen has previously been used for single character Ott et al. (2020; 2022a;b) and sequence Ott et al. (2022c) classification. However, the accuracy of previous online HWR methods is limited due to the limited size of datasets as recording of larger amounts of data is time consuming. A possible solution is to combine datasets of different modalities using common representation learning to increase the generalizability. In this work, we combine offline HWR from generated images (i.e., OCR) and online HWR from sensor-enhanced pens by learning a common representation between both modalities. The aim is to integrate information of OCR, i.e., typeface, cursive or printed writing, and font thickness, into the online HWR task, i.e., writing speed and direction Vinciarelli & Perrone (2003).

**Our Contribution.**   Models that use rich data (e.g., images) usually outperform those that use a less rich modality (e.g., time-series). We therefore propose to train a shared representation using the triplet loss between pairs of image and time-series data to learn a common representation between both modality embeddings (cf. Figure 1). This allows to improve the accuracy for single-modal inference in the main task. We prove the efficacy of our DML-based triplet loss for CRL both with simulated data and in a real-world application. More specifically, our proposed CRL technique 1) improves the multivariate time-series (MTS) classification accuracy and convergence, 2) results in a small MTS-only network independent from the image modality while allowing for fast inference, and 3) has better generalizability and adaptability Huang et al. (2020). Code and datasets are available upon publication.[1]

The paper is organized as follows. Section 2 discusses related work followed by the mathematical foundation of our method in Section 3. The experimental setup is described in Section 4 and the results are discussed in Section 5.

## 2   Related Work

In this section, we discuss related work, in particular approaches for learning a common representation from different modalities (in Section 2.1) and DML (in Section 2.2) to minimize the distance between feature embeddings.

### 2.1   Cross-Modal Representation Learning

For traditional methods that learn a common representation, a cross-modal similarity for the retrieval can be calculated with linear projections as basic models Rasiwasia et al. (2010). However, cross-modal correlation is highly complex,

---

[1]Code and datasets: https://www.anonymous-submission.org

and hence, recent methods are based on a *modal-sharing network* to jointly transfer non-linear knowledge from a single modality to all modalities Wei et al. (2016). Huang et al. (2020) use a *cross-modal network* between different modalities (image to video, text, audio and 3D models) and a *single-modal network* (shared features between images of source and target domains). They use two convolutional layers (similar to our proposed architecture) that allows the model to adapt more trainable parameters. However, while their auxiliary network uses the same modality, our auxiliary network is based on another modality. Lee et al. (2020) learn a common embedding between video frames and audio signals with graph clusters, but at inference both modalities must be available. Sarafianos et al. (2019) proposed an image-text modality adversarial matching approach that learns modality-invariant feature representations, but their projection loss is used for learning discriminative image-text embeddings only. Hafner et al. (2022) propose a model for single-modal inference. However, they use image and depth modalities for person re-identification without a time-series component, which makes the problem considerably different. Lim et al. (2019) handled multisensory modalities for 3D models only.

## 2.2 Deep Metric Learning

Networks trained for the classification task can produce useful feature embeddings with efficient runtime complexity $\mathcal{O}(NC)$ per epoch, where $N$ is the number of training samples and $C$ the number of classes. The classical cross-entropy (CE) loss, however, is not useful for DML as it ignores how close each point is to its class centroid (or how far apart from other class centroids). The *pairwise contrastive loss* Chopra et al. (2005) minimizes the distance between feature embedding pairs of the same class and maximizes the distance between feature embedding pairs of different classes dependent on a margin parameter. The issue is that the optimization of positive pairs is independent from negative pairs, but the optimization should force the distance between positive pairs to be smaller than negative pairs Do et al. (2019).

The *triplet loss* Yoshida et al. (2019) addresses this by defining an anchor and a positive as well as a negative point, and forces the positive pair distance to be smaller than the negative pair distance by a certain margin. The runtime complexity of the triplet loss is $\mathcal{O}(N^3/C)$, and can be computationally challenging for large training sets. Hence, several works exist to reduce this complexity such as hard or semi-hard triplet mining Schroff et al. (2015), or smart triplet mining Harwood et al. (2017). Often, data is evolving over time, and hence, Semedo & Magalhães (2020) proposed a formulation of the triplet loss where the traditional static *margin* is superseded by a temporally adaptive maximum margin function. While Zeng et al. (2017); Li et al. (2021) combine the triplet loss with the CE loss, Guo et al. (2019) use a triplet selection with $L_2$-normalization for language modeling, but considered all negative pairs for triplet selection with fixed similarity intensity parameter. For our experiments, we use a triplet loss with a dynamic margin together with a novel word level triplet selection. The DeepTripletNN Zeng et al. (2020) also uses the triplet loss on embeddings between an anchor from audio data and positive and negative samples from visual data, and the cosine similarity for the final representation comparison. CrossATNet Chaudhuri et al. (2020), another triplet loss-based method which uses single class labels, defines class sketch instances as anchor, the same class image instance as positive sample and a different class image instance as negative sample. While the previous methods are based on a triplet selection method using single label classification, we are – to the best of our knowledge – the first to propose the triplet loss for sequence-based classification (i.e., words).

Most of the related work uses the Euclidean metric as distance loss, although the triplet loss can be defined based on any other (sub-)differentiable distance metric. Wan & Zou (2021) proposed a method for offline signature verification based on a dual triplet loss that uses the Euclidean space to project an input image to an embedding function. While Rantzsch et al. (2016) use the Euclidean metric to learn the distance between feature embeddings, Zeng et al. (2017) use the Cosine similarity. Hermans et al. (2017) state that using the *non-squared* Euclidean distance is more stable, while the *squared* distance made the optimization more prone to collapsing. Recent methods extend the canonical correlation analysis (CCA) Ranjan et al. (2015) that learns linear projection matrices by maximizing pairwise correlation of cross-modal data. To share information between the same modality (i.e., images), typically the maximum mean discrepancy (MMD) Long et al. (2015) is minimized.

# 3 Methodology

We define the problem of common representation learning and present DML loss functions in Section 3.1. In Section 3.2 we propose the triplet loss for cross-modal learning.

## 3.1 Common Representation Learning

A multivariate time-series (MTS) $\mathbf{U} = \{\mathbf{u}_1, \ldots, \mathbf{u}_m\} \in \mathbb{R}^{m \times l}$ is an ordered sequence of $l \in \mathbb{N}$ streams with $\mathbf{u}_i = (u_{i,1}, \ldots, u_{i,l}), i \in \{1, \ldots, m\}$, where $m \in \mathbb{N}$ is the length of the time-series. The MTS training set is a subset of the array $\mathcal{U} = \{\mathbf{U}_1, \ldots, \mathbf{U}_{n_U}\} \in \mathbb{R}^{n_U \times m \times l}$, where $n_U$ is the number of time-series. Let $\mathbf{X} \in \mathbb{R}^{o \times p}$ with entries $x_{i,j} \in [0, 255]$ represent an image from the image training set. The image training set is a subset of the array $\mathcal{X} = \{\mathbf{X}_1, \ldots, \mathbf{X}_{n_X}\} \in \mathbb{R}^{n_X \times o \times p}$, where $n_X$ is the number of time-series. The aim of joint MTS and image classification tasks is to predict an unknown class label $v \in \Omega$ for single class prediction or $\mathbf{v} \in \Omega$ for sequence prediction for a given MTS or image (see also Section 4.2). The time-series samples denote the main training data, while the image samples represent the privileged information that is not used for inference. In addition to good prediction performance, the goal is to learn representative embeddings $f_c(\mathbf{U})$ and $f_c(\mathbf{X}) \in \mathbb{R}^{q \times w}$ to map MTS and image data into a feature space $\mathbb{R}^{q \times w}$, where $f_c$ is the output of the convolutional layer(s) $c \in \mathbb{N}$ of the latent representation.

We force the embedding to live on the $q \times w$-dimensional hypersphere by using a $\mathtt{Softmax}$ attention, i.e., $||f_c(\mathbf{U})||_2 = 1$ and $||f_c(\mathbf{X})||_2 = 1 \,\forall c$ (see (Weinberger et al., 2005)). In order to obtain a small distance between the embeddings $f_c(\mathbf{U})$ and $f_c(\mathbf{X})$, we minimize DML functions $\mathcal{L}_{\mathrm{DML}}(f_c(\mathbf{X}), f_c(\mathbf{U}))$. Well-known DML metrics are the distance-based mean squared error (MSE) $\mathcal{L}_{\mathrm{MSE}}$, the spatio-temporal cosine similarity (CS) $\mathcal{L}_{\mathrm{CS}}$, the Pearson correlation (PC) $\mathcal{L}_{\mathrm{PC}}$, or the distribution-based Kullback-Leibler (KL) divergence $\mathcal{L}_{\mathrm{KL}}$. In our experiments, we additionally evaluate the kernalized maximum mean discrepancy (kMMD) $\mathcal{L}_{\mathrm{kMMD}}$, Bray Curtis (BC) $\mathcal{L}_{\mathrm{BC}}$, and Poisson $\mathcal{L}_{\mathrm{PO}}$ losses. We study their performance in Section 5. A combination of classification and CRL losses can be realized by dynamic weight averaging Liu et al. (2019) as a multi-task learning approach that performs dynamic task weighting over time (see Appendix A.1).

## 3.2 Triplet Loss

While the training with the previous loss functions uses inputs where the image and MTS have the same label, pairs with similar but different labels can improve the training process. This can be achieved using the triplet loss Schroff et al. (2015) which enforces a margin between pairs of image and MTS data with the same identity to all other different identities. As a consequence, the convolutional output for one and the same label lives on a manifold, while still enforcing the distance and thus discriminability to other identities. We therefore seek to ensure that the embedding of the MTS $\mathbf{U}_i^a$ (*anchor*) of a specific label is closer to the embedding of the image $\mathbf{X}_i^p$ (*positive*) of the same label than it is to the embedding of any image $\mathbf{X}_i^n$ (*negative*) of another label (see Figure 2). Thus, we want the following inequality to hold for all training samples $(f_c(\mathbf{U}_i^a), f_c(\mathbf{X}_i^p), f_c(\mathbf{X}_i^n)) \in \Phi$:

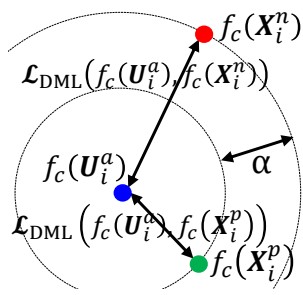

Figure 2: Triplet pair.

$$\mathcal{L}_{\mathrm{DML}}\big(f_c(\mathbf{U}_i^a), f_c(\mathbf{X}_i^p)\big) + \alpha < \mathcal{L}_{\mathrm{DML}}\big(f_c(\mathbf{U}_i^a), f_c(\mathbf{X}_i^n)\big), \tag{1}$$

where $\mathcal{L}_{\mathrm{DML}}\big(f_c(\mathbf{X}), f_c(\mathbf{U})\big)$ is a DML loss, $\alpha$ is a margin between positive and negative pairs, and $\Phi$ is the set of all possible triplets in the training set. Based on (1), we can formulate a differentiable loss function that we can use for optimization:

$$\mathcal{L}_{\mathrm{trpl,c}}(\mathbf{U}^a, \mathbf{X}^p, \mathbf{X}^n) = \sum_{i=1}^{N} \max\Big[\mathcal{L}_{\mathrm{DML}}\big(f_c(\mathbf{U}_i^a), f_c(\mathbf{X}_i^p)\big) - \mathcal{L}_{\mathrm{DML}}\big(f_c(\mathbf{U}_i^a), f_c(\mathbf{X}_i^n)\big) + \alpha, 0\Big], \tag{2}$$

where $c \in \mathbb{N}$.[2] Selecting negative samples that are too close to the anchor (in relation to the positive sample) can cause slow training convergence. Hence, triplet selection must be handled carefully and application-specific Do et al. (2019). We choose negative samples based on the class distance (single labels) and on the Edit distance (sequence labels), see Section 4.2.

---

[2]To have a larger number of trainable parameters in the latent representation with a greater depth, we evaluate one and two stacked convolutional layers, each trained with a shared loss $\mathcal{L}_{\mathrm{trpl,c}}$.

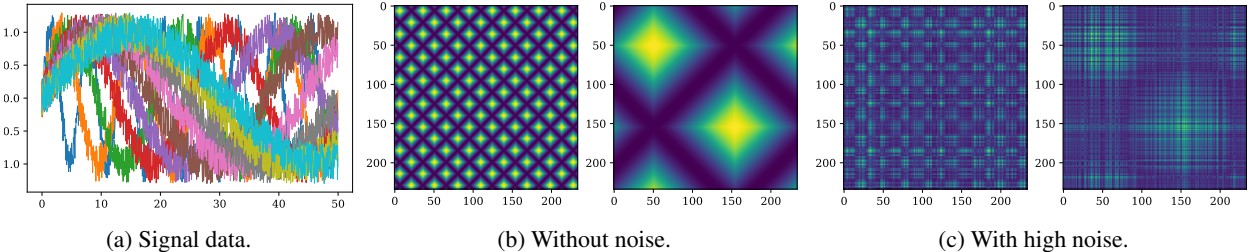

|              |                  |                    |
|:------------:|:----------------:|:------------------:|
| (a) Signal data. | (b) Without noise. | (c) With high noise. |

Figure 3: Synthetic signal data (a) for 10 classes, and image data (b-c) for classes 0 (left) and 6 (right).

# 4 Experiments

We now demonstrate the efficacy of our proposal. In Section 4.1 we generate sinusoidal time-series with introduced noise (main task) and compute the corresponding Gramian angular summation field (GASF) with different noise parameters (auxiliary task), see Figure 1. In Section 4.2 we combine online (inertial sensor signals, main task) and offline data (visual representations, auxiliary task) for handwriting recognition (HWR) with sensor-enhanced pens. This task is particularly challenging due to different data representations based on images and MTS data. For both applications, our approach allows to only use the main modality (MTS) for inference. We further analyze and evaluate different DML functions to minimize the distance between the learned embeddings.

## 4.1 Cross-Modal Learning on Synthetic Data

We first investigate the influence of the triplet loss for cross-modal learning between synthetic time-series and image-based data. For this, we generate signal data of 1,000 timesteps with different frequencies for 10 classes (see Figure 3a) and add noise from a continuous uniform distribution $U(a, b)$ for $a = 0$ and $b = 0.3$. We use a recurrent CNN with the CE loss to classify these signals. From each signal without noise, we generate a GASF Wang & Oates (2015). For classes with high frequencies, this results in a fine-grained pattern, and for low frequencies in a coarse-grained pattern. We generate GASFs with different added noise between $b = 0$ (Figure 3b) and $b = 1.95$ (Figure 3c). A small CNN classifies these images with the CE loss. To combine both networks, we train each signal-image pair with the triplet loss. As the frequency of the sinusoidal signal is closer for more similar class labels, the distance in the manifold embedding should also be closer. For each batch, we select negative sample pairs for samples with the class label $CL = 1 + \lfloor \frac{\max_e - e - 1}{25} \rfloor$ as lower bound for current epoch $e$ and maximum epoch $\max_e$. We set the margin $\alpha$ in the triplet loss separately for each batch such that $\alpha = \beta \cdot (CL_p - CL_n)$ depends on the positive $CL_p$ and negative $CL_n$ class labels of the batch and is in the range $[1, 5]$ with $\beta = 0.1$. The batch size is 100 and $\max_e = 100$. Appendix A.2 provides further details. This combination of the CE loss with the triplet loss can lead to a mutual improvement of the utilization of the classification task and embedding learning.

## 4.2 Cross-Modal Learning for HWR

**Method Overview.** Figure 4 gives a method overview. The main task is online HWR to classify words written with a sensor-enhanced pen and represented by MTS of the different pen sensors. To improve the classification task with a better generalizability, the auxiliary network performs offline HWR based on an image input. We pre-train ScrabbleGAN Fogel et al. (2020) on the IAM-OffDB Liwicki & Bunke (2005) dataset and for all MTS word labels generate the corresponding image as the positive MTS-image pair. Each MTS and each image is associated with $\mathbf{v}$, a sequence of $L$ class labels from a pre-defined label set $\Omega$ with $K$ classes. For our classification task, $\mathbf{v} \in \Omega^L$ describes words. The MTS training set is a subset of the array $\mathcal{U}$ with labels $\mathcal{V}_U = \{\mathbf{v}_1, \ldots, \mathbf{v}_{n_U}\} \in \Omega^{n_U \times L}$. The image training set is a subset of the array $\mathcal{X}$, and the corresponding labels $\mathcal{V}_X = \{\mathbf{v}_1, \ldots, \mathbf{v}_{n_X}\} \in \Omega^{n_X \times L}$. Offline HWR techniques are based on Inception, ResNet34, or GTR Yousef et al. (2018) modules. The online method is improved by sharing layers with a common representation by minimizing the distance of the feature embedding of the convolutional layers $c \in \{1, 2\}$ (integrated in both networks) with a shared loss $\mathcal{L}_{\text{shared},c}$. We set the embedding size $\mathbb{R}^{q \times w}$ to $400 \times 200$. Both networks are trained with the connectionist temporal classification (CTC) Graves

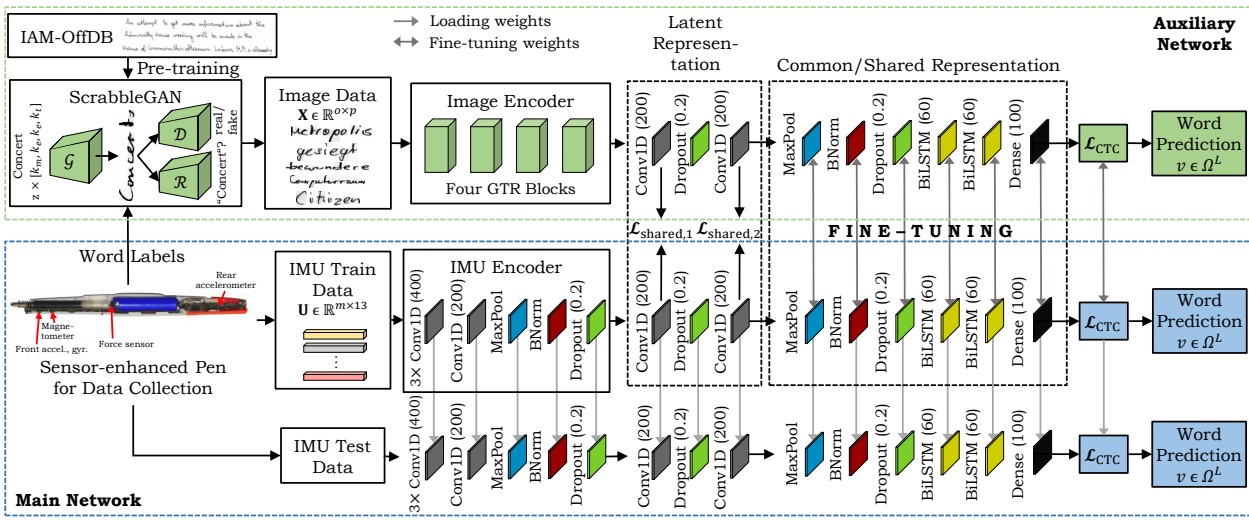

Figure 4: **Detailed method overview:** The middle pipeline consists of data recording with a sensor-enhanced pen, feature extraction of inertial MTS data, and word classification with CTC. We generate image data with the pre-trained ScrabbleGAN for corresponding word labels. The top pipeline (four GTR blocks) extracts features from images. The distances of the embeddings are minimized with the triplet loss and DML functions. The classification network with two BiLSTM layers are fine-tuned for the OnHW task for a common representation.

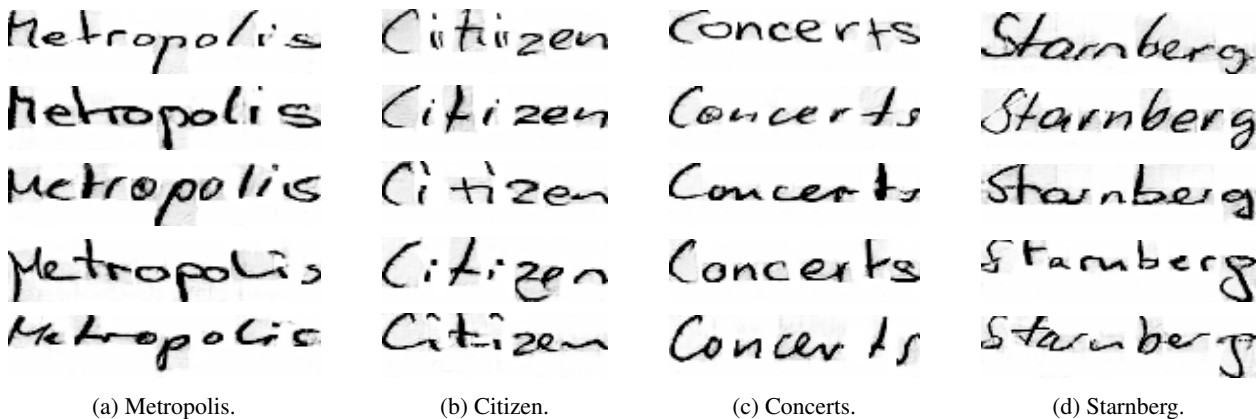

| (a) Metropolis. | (b) Citizen. | (c) Concerts. | (d) Starnberg. |

Figure 5: Overview of four generated words with ScrabbleGAN Fogel et al. (2020) with various text styles.

et al. (2009) loss $\mathcal{L}_{\text{CTC}}$ to avoid pre-segmentation of the training samples by transforming the network outputs into a conditional probability distribution over label sequences.

**Datasets for Online HWR.** We make use of two word datasets proposed in Ott et al. (2022c). These datasets are recorded with a sensor-enhanced pen that uses two accelerometers (3 axes each), one gyroscope (3 axes), one magnetometer (3 axes), and one force sensor at 100 Hz Ott et al. (2020; 2022b). One sample of size $m \times l$ represents an MTS of a written word of $m$ timesteps from $l = 13$ sensor channels. One word is a sequence of small or capital characters (52 classes) or with mutated vowels (59 classes). The *OnHW-words500* dataset contains 25,218 samples where each of the 53 writers contributed the same 500 words. The *OnHW-wordsRandom* dataset contains 14,641 randomly selected words from 54 writers. For both datasets, 80/20 train/validation splits are available for writer-(in)dependent (WD/WI) tasks. We transform (zero padding, interpolation) all samples to 800 timesteps.

**Image Generation for Offline HWR.** In order to couple the online MTS data with offline image data, we use a generative adversarial network (GAN) to generate arbitrarily many images. ScrabbleGAN Fogel et al. (2020) is a state-of-the-art semi-supervised approach that consists of a generator $\mathcal{G}$ that generates images of words with arbitrary length

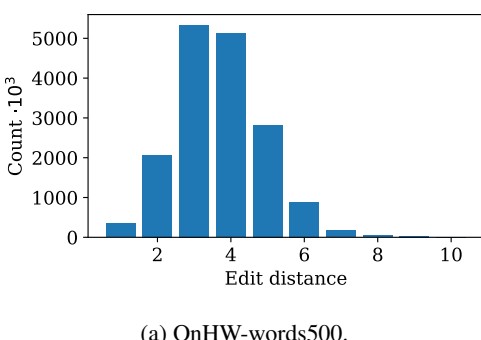

(a) OnHW-words500.

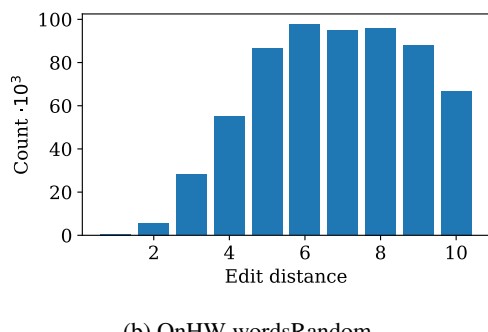

(b) OnHW-wordsRandom.

Figure 6: Number image-MTS pairs dependent on mismatches.

from an input word label, a discriminator $\mathcal{D}$, and a recognizer $\mathcal{R}$ promoting style and data fidelity. For the generator, four character filters ($k_m$, $k_e$, $k_e$ and $k_t$) are concatenated, multiplied by a noise vector and fed into a class-conditioned generator. This allows for adjacent characters to interact, e.g., enabling cursive text. We train ScrabbleGAN with the IAM-OffDB Liwicki & Bunke (2005) dataset and generate three different datasets. Exemplary images are shown in Figure 5. First, we generate 2 million images randomly selected from a large lexicon (*OffHW-German*), and pre-train the offline HWR architectures. Second, we generate 100,000 images based on the same word labels for each of the OnHW-words500 and OnHW-wordsRandom datasets (*OffHW-*[*words500*, *wordsRandom*]), and fine-tune the offline HWR architectures.

**Methods for Offline HWR.** OrigamiNet Yousef & Bishop (2020) is a state-of-the-art multi-line recognition method using only unsegmented image and text pairs. An overview of offline HWR methods is given in Appendix A.3. Similar to OrigamiNet, our offline method is based on different encoder architectures with one or two additional 1D convolutional layers (each with filter size 200, `Softmax` activation Zeng et al. (2017)) with 20% dropout for the latent representation, and a common representation decoder with BiLSTMs. For the encoder, we make use of Inception modules from GoogLeNet, the ResNet34 architecture, and re-implement the newly proposed gated, fully convolutional method gated text recognizer (GTR) Yousef et al. (2018). See Appendix A.4 for detailed information on the architectures. We train the networks on the generated OffHW-German dataset for 10 epochs, and fine-tune on the OffHW-[500, wordsRandom] datasets for 15 epochs. For comparison with state-of-the-art techniques, we train OrigamiNet and compare with IAM-OffDB. For OrigamiNet, we apply interline spacing reduction via seam carving Avidan & Shamir (2007), resizing the images to 50% height, and random projective (rotating and resizing lines) and random elastic transform Wigington et al. (2017). We augment the OffHW-German dataset with random width resizing and apply no augmentation for the OffHW-[words500, wordsRandom] datasets for fine-tuning.

**Offline/Online Common Representation Learning.** Our architecture for online HWR is based on Ott et al. (2022c). The encoder extracts features of the inertial data and consists of three convolutional layers (each with filter size 400, `ReLU` activation) and one convolutional layer (filter size 200, `ReLU` activation), a max pooling, batch normalization and a 20% dropout layer. As for the offline architecture, the network then learns a latent representation with one or two convolutional layers (each with filter size 200, `Softmax` activation) with 20% dropout and the same CRL decoder. The output of the convolutional layers of the latent representation are minimized with the $\mathcal{L}_{\text{shared,c}}$ loss. The layers of the common representation are fine-tuned based on the pre-trained weights of the offline technique. Here, two BiLSTM layers with 60 units each and `ReLU` activation extract the temporal context of the feature embedding. As for the baseline classifier, we train for 1,000 epochs. For evaluation, the main MTS network is independent of the image auxiliary network by using only the weights of the main network.

**Triplet Selection.** To ensure (fast) convergence, it is crucial to select triplets that violate the constraint from Equation 1. Typically, it is infeasible to compute the loss for all triplet pairs or this leads to poor training performance as poorly chosen pairs dominate hard ones. This requires an elaborate triplet selection Do et al. (2019). We use the Edit distance (ED) to define the identity and select triplets. The ED is the minimum number of substitutions $S$, insertions $I$ and deletions $D$ required to change the sequences $\mathbf{h} = (h_1, \ldots, h_r)$ into $\mathbf{g} = (g_1, \ldots, g_t)$ with length $r$ and $t$, respec-

tively. We define two sequences with an ED of 0 as positive pair, and with an ED larger than 0 as negative pair. Based on preliminary experiments, we use only substitutions for triplet selection that lead to a higher accuracy compared to additional insertions and deletions (whereas these would also change the length difference of image and MTS pairs). We constrain $p - m/2$, the difference in pixels $p$ of the images and half the number of timesteps of the MTS, to be maximal $\pm 20$. The goal is a small distance for positive pairs, and a large distance for negative pairs that increases with a larger ED (between 1 and 10). And despite a limited number of word labels, there still exist a large number of image-MTS pairs per word label for every possible ED (see Figure 6). For each batch, we search in a dictionary of negative sample pairs for samples with $ED = 1 + \lfloor \frac{\max_e - e - 1}{100} \rfloor$ as lower bound for the current epoch $e$ and maximal epochs $\max_e$. For every label we randomly pick one image. We let the margin $\alpha$ in the triplet loss vary for each batch such that $\alpha = \beta \cdot \overline{ED}$ is depending on the mean ED of the batch and is in the range $[1, 11]$ with $\beta = 10^{-3}$ for MSE, $\beta = 0.1$ for CS and PC, and $\beta = 1$ for KL. The batch size is 100 and $\max_e = 1,000$.

## 5 Experimental Results

**Hardware and Training Setup.** For all experiments we use Nvidia Tesla V100-SXM2 GPUs with 32 GB VRAM equipped with Core Xeon CPUs and 192 GB RAM. We use the vanilla Adam optimizer with a learning rate of $10^{-4}$.

### 5.1 Evaluation of Synthetic Data

We train the time-series (TS) model 18 times with noise $b = 0.3$, and the combined model with the triplet loss for all 40 noise combinations $(b \in \{0, \ldots, 1.95\})$ with different DML functions. Figure 7 shows the validation accuracy averaged over all trainings as well as the combined cases separately for noise $b < 0.2$ and noise $0.2 \leq b < 2.0$ (for the $\mathcal{L}_{CS}$ loss). The accuracy of the models that use only images and in combination with MTS during inference reach an accuracy of 99.7% (which can be seen as an unreachable upper bound for the TS-only models). The triplet loss improves the final TS baseline accuracy from 92.5% to 95.36% (averaged over all combinations) while combining TS and image data leads to a faster convergence. Conceptually similar to Long et al. (2015), we use the $\mathcal{L}_{kMMD}$ loss which yields 95.83% accu-

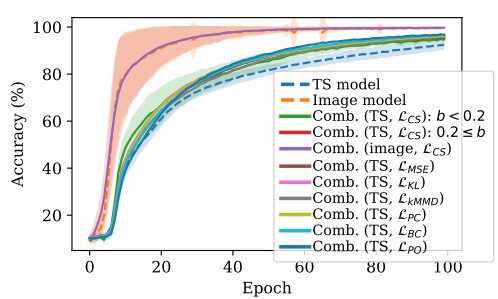

Figure 7: Comparison of single- and cross-modal CRL.

racy. The $\mathcal{L}_{PC}$ (96.03%), $\mathcal{L}_{KL}$ (96.22%), $\mathcal{L}_{MSE}$ (96.25%), $\mathcal{L}_{BC}$ (96.62%), and $\mathcal{L}_{PO}$ (96.76%) loss functions can further improve the accuracy. We conclude that the triplet loss can be successfully used for cross-modal learning by utilizing negative identities.

### 5.2 Evaluation of HWR

**Evaluation Metrics.** A metric for sequence evaluation is the character error rate (CER) defined as $\text{CER} = \frac{S_c + I_c + D_c}{N_c}$ as the Edit distance (the sum of character substitutions $S_c$, insertions $I_c$ and deletions $D_c$) divided by the total number of characters in the set $N_c$. Similarly, the word error rate (WER) is defined as $\text{WER} = \frac{S_w + I_w + D_w}{N_w}$ computed with word operations $S_w$, $I_w$ and $D_w$, and number of words in the set $N_w$.

**Evaluation of Offline HWR Methods.** All our models yield low error rates on the generated OffHW-German dataset. Our approach with GTR blocks outperforms (0.24% to 0.44% CER) the models with Inception (1.27% CER) and ResNet (1.24% CER). OrigamiNet achieves the lowest error rates of 1.50% WER and 0.11% CER. Four GTR blocks yield the best results at a significantly lower training time compared to six or eight blocks. We fine-tune the model with four GTR blocks for one and two convolutional layers and achieve notably low error rates between 0.22% to 0.76% CER, and between 0.85% to 2.95% WER on the OffHW-[words500, wordsRandom] datasets. For more results, see Appendix A.5.

**Evaluation of CRL Feature Embeddings.** Table 1 shows the feature embeddings for image $f_2(\mathbf{X}_i)$ and MTS data $f_2(\mathbf{U}_i)$ of the *positive* sample `Export` and the two *negative* samples `Expert` ($ED = 1$) and `Import` ($ED = 2$) based on four DML loss functions. The pattern of characters are similar as the words differ only in the fourth letter.

Table 1: Feature embeddings $f_c(\mathbf{X}_i)$ and $f_c(\mathbf{U}_i)$ of exemplary image $\mathbf{X}_i$ and MTS $\mathbf{U}_i$ data of the convolutional layer $c = \text{conv}_2$ for different deep metric learning functions for positive pairs ($ED = 0$) and negative pairs ($ED > 0$) trained with the triplet loss. The feature embeddings are similar in the red box (character x) or blue box (character p) for $f_2(\mathbf{X}_i)$, or the last pixels (character t) of $f_2(\mathbf{U}_i)$ for $\mathcal{L}_{\text{PC}}$ marked green.

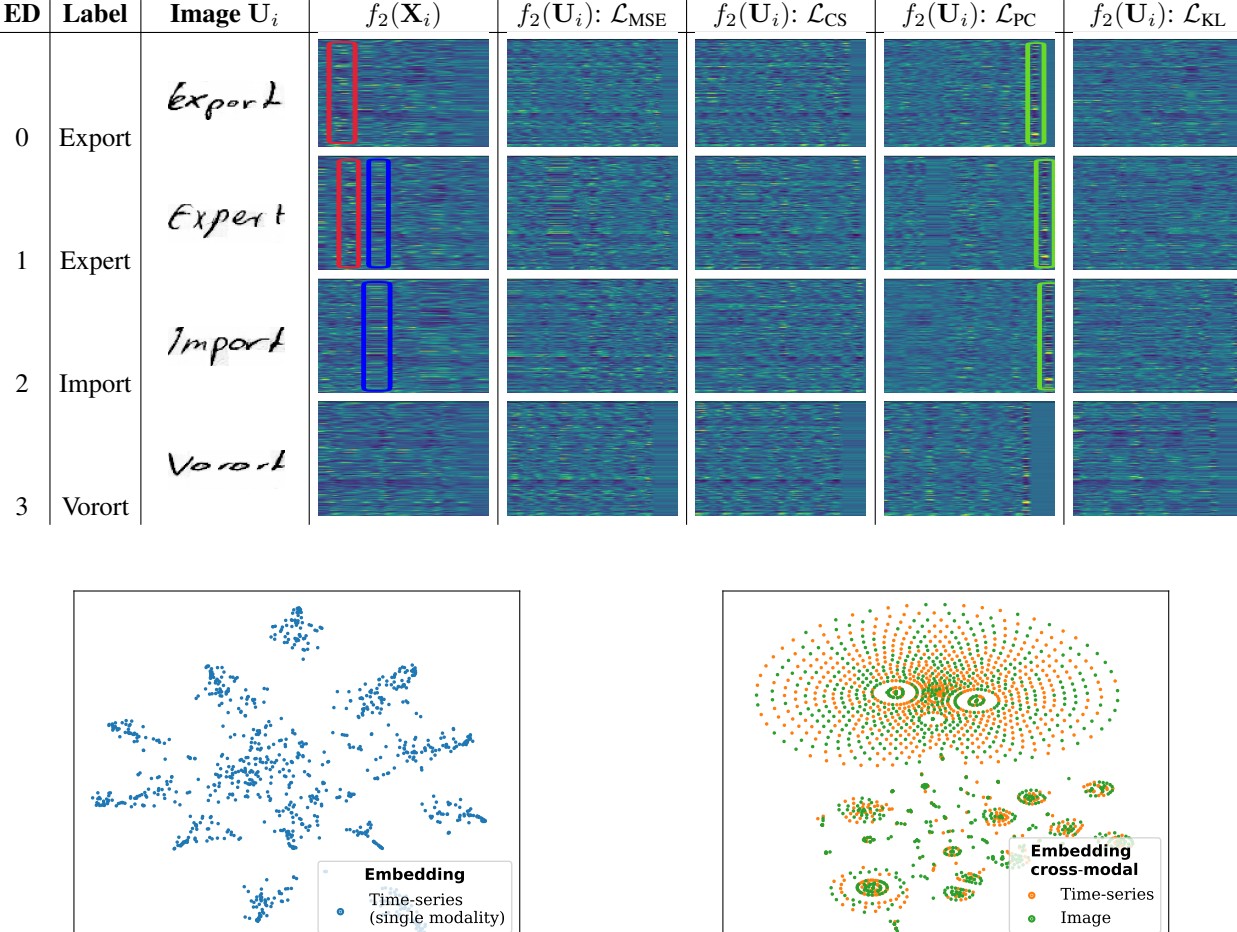

| ED | Label | Image $\mathbf{U}_i$ | $f_2(\mathbf{X}_i)$ | $f_2(\mathbf{U}_i)$: $\mathcal{L}_{\text{MSE}}$ | $f_2(\mathbf{U}_i)$: $\mathcal{L}_{\text{CS}}$ | $f_2(\mathbf{U}_i)$: $\mathcal{L}_{\text{PC}}$ | $f_2(\mathbf{U}_i)$: $\mathcal{L}_{\text{KL}}$ |
|----|-------|------|------|------|------|------|------|
| 0 | Export | | | | | | |
| 1 | Expert | | | | | | |
| 2 | Import | | | | | | |
| 3 | Vorort | | | | | | |

(a) Feature embedding of IMU samples for the single modalitiy network.

(b) Feature embeddings of IMU and image samples for the cross-modal network.

Figure 8: Plot of $400 \times 200$ feature embeddings of image and IMU modalities with t-SNE.

In contrast, Import has a different feature embedding as the replacement of E with I and x with m leads to a higher feature distance in the embedding hypersphere. Note that image and MTS data can vary in length for $ED > 0$. Figure 8 shows the feature embeddings of the output of the convolutional layers ($c = 1$) processed with t-SNE van der Maaten & Hinton (2008). Figure 8a visualizes the MTS embeddings $f_1(\mathbf{U}_i)$ of the single modal network, and Figure 8b visualizes the MTS and image embeddings, $f_1(\mathbf{U}_i)$ and $f_1(\mathbf{X}_i)$ respectively, in a cross-modal setup. While the embedding of the single modal network is unstructured, the embeddings of the cross-modal network are structured (distance of samples visualizes the Edit distance between words).

**Evaluation of Cross-Modal CRL.** Table 2 gives an overview of CRL (for $c = 1$). The first row are baseline results by Ott et al. (2022c): 13.04% CER on OnHW-words500 (WD) and 6.75% CER on OnHW-wordsRandom (WD) with mutated vowels (MV). Compared to various time-series classification techniques, their benchmark results showed superior performance of CNN+BiLSTMs on these OnHW recognition tasks. In general, the word error rate (WER) can vary for a similar character error rate (CER). The reason is that a change of one character of a correctly classified

Table 2: Evaluation results (WER and CER in %) average over five splits of the baseline MTS-only technique and our cross-modal learning technique for the inertial-based OnHW datasets Ott et al. (2022c) with and without mutated vowels (MV) for one convolutional layer $c = 1$.

| Method | OnHW-words500 | | | | OnHW-wordsRandom | | | |
| | WD | | WI | | WD | | WI | |
| | WER | CER | WER | CER | WER | CER | WER | CER |
|---|---|---|---|---|---|---|---|---|
| $\mathcal{L}_{\text{CTC}}$, w/ MV | 42.81 | 13.04 | 60.47 | 28.30 | 37.13 | 6.75 | 83.28 | 35.90 |
| $\mathcal{L}_{\text{CTC}}$, w/o MV | 42.77 | 13.44 | 59.82 | 28.54 | 38.02 | 7.81 | 83.54 | 36.51 |
| $\mathcal{L}_{\text{MSE}}$ | 40.76 | 12.71 | **55.54** | 25.97 | 37.31 | 7.01 | 82.25 | 33.85 |
| $\mathcal{L}_{\text{CS}}$ | 38.62 | 11.55 | 56.37 | **25.90** | 38.85 | 7.35 | 82.48 | 35.67 |
| $\mathcal{L}_{\text{PC}}$ | 39.09 | 11.69 | 57.90 | 27.23 | 38.46 | 7.15 | 82.71 | 35.13 |
| $\mathcal{L}_{\text{KL}}$ | 38.36 | 11.28 | 60.23 | 27.99 | 38.76 | 7.49 | **81.07** | 33.96 |
| $\mathcal{L}_{\text{trpl,1}}(\mathcal{L}_{\text{MSE}})$ | 42.95 | 14.13 | 56.48 | 26.66 | 37.66 | 7.04 | 81.64 | 34.39 |
| $\mathcal{L}_{\text{trpl,1}}(\mathcal{L}_{\text{CS}})$ | 38.01 | 11.29 | 58.50 | 27.10 | **37.12** | **6.98** | 82.71 | **33.09** |
| $\mathcal{L}_{\text{trpl,1}}(\mathcal{L}_{\text{PC}})$ | 40.43 | 12.41 | 58.20 | 27.48 | 37.40 | 7.01 | 81.90 | 33.89 |
| $\mathcal{L}_{\text{trpl,1}}(\mathcal{L}_{\text{KL}})$ | **37.55** | **11.21** | 63.52 | 30.52 | 38.39 | 7.36 | 83.18 | 35.21 |

word leads to a large change in the WER, while the change of the CER is marginal. We compare to results without MV as ScrabbleGAN is pretrained on IAM-OffDB that does not contain MV, and hence, such words cannot be generated. Here, the error rates are slightly higher for both datasets. As expected, cross-modal learning improves the baseline results up to 11.28% CER on the OnHW-words500 WD dataset and up to 7.01% CER on the OnHW-wordsRandom WD dataset. With triplet loss, $\mathcal{L}_{\text{CS}}$ outperforms other metrics on the OnHW-wordsRandom dataset, but is inconsistent on the OnHW-words500 dataset. The importance of the triplet loss is more significant for one convolutional layer ($c = 1$) than for two convolutional layers ($c = 2$), see Appendix A.5. Further, training with kMMD (implemented as in (Long et al., 2015)) does not yield reasonable results. We assume that this metric cannot make use of the important time component in the HWR application.

# 6 Conclusion

We evaluated DML-based triplet loss functions for CRL between image and time-series modalities with class label specific triplet selection. On synthetic data as well as on different HWR datasets, our method yields notable accuracy improvements for the main time-series classification task and can be decoupled from the auxiliary image classification task at inference time. Our cross-modal triplet selection further yields a faster training convergence with better generalization on the main task.

**Broader Impact Statement**

While research for offline handwriting recognition (HWR) is very advanced (an overview is proposed in the appendix), research for online HWR from sensor-enhanced only emerged in 2019. Hence, the methodological research currently does not meet the requirements for real-world applications. Handwriting is still important in different fields, in particular graphomotoric. The visual feedback provided by the pen helps young students to learn a new language. A well-known bottleneck for many machine learning algorithms is their requirement for large amounts of datasets, while data recording of handwriting data is time consuming. This paper extends the online HWR dataset with generated images from offline handwriting, and closes the gap between offline and online HWR by using offline HWR as auxiliary task by learning with privileged information. One downside of training the offline architecture (consisting of GTR blocks) is its long training time. But as this model is not required at inference time, processing the time-series is still fast. The common representation between both modalities (image and time-series) is achieved by using the triplet loss and a sample selection depending on the Edit distance. This approach is important in future applications of sequence-based classification as the triplet loss may also evolve for language processing applications as strong as in typically applied fields such as image recognition. Ethical statement about collection consent and personal information: For data recording, the consent of all participants was collected. The datasets only contain the raw data from the sensor-enhanced pen, and for statistics the age and gender of the participants and their handedness. The dataset is

fully pseudonymized by assigning an ID to every participant. The dataset does not contain any offensive content. The approach proposed in this paper, in particular used for the application of online handwriting recognition from sensor-enhanced pens, does not (1) facilitate injury to living beings, (2) raise safety or security concerns due to the anonymity of the data, (3) raise human rights concerns, (4) have a detrimental effect on people's livelihood, (5) develop harmul forms of surveillance as the data is pseudonymized, (6) damage the environment, and (7) deceive people in ways that cause harm.

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

# A Appendices

We present the multi-task learning technique in Section A.1, and show more details on learning with the triplet loss on synthetically generated signal and image data in Section A.2. We present an method overview for offline handwriting recognition (HWR) in Section A.3, and propose more details of our architectures in Section A.4. Section A.5 presents results of representation learning for HWR.

## A.1 Multi-Task Learning (MTL)

We simultaneously train the $\mathcal{L}_{\text{CTC}}$ loss for sequence classification combined with one or two shared losses $\mathcal{L}_{\text{shared,1}}$ and $\mathcal{L}_{\text{shared,2}}$ for common representation learning (CRL). As both losses are in different ranges, the naive weighting

$$\mathcal{L}_{\text{total}} = \sum_{i=1}^{|T|} \omega_i \mathcal{L}_i, \tag{3}$$

with pre-specified, constant weights $\omega_i = 1, \forall i \in \{1, \dots, |T|\}$ can harm the training process. Hence, we apply dynamic weight average (DWA) Liu et al. (2019) as an MTL approach that performs dynamic task weighting over time (i.e., after each batch).

## A.2 Training Synthetic Data with the Triplet Loss

**Signal and Image Generation.** We combine the networks for both, signal and image classification, to improve the classification accuracy over each single-modal network. The aim is to show that the triplet loss can be used for such a cross-modal setting in the field of common representation learning. Hence, we generate synthetic data where the image data contains information of the signal data. We generate signal data $\mathbf{x}$ with $x_{i,k} = \sin\left(0.05 \cdot \frac{t_i}{k}\right)$ for all $t_i \in \{1, \dots, 1,000\}$ where $t_i$ is the timestep of the signal. The frequency of the signal is dependent on the class label $k$. We generate signal data for 10 classes (see Figure 9a). We add noise from a continuous uniform distribution $U(a, b)$ for $a = 0$ and $b = 0.3$ (see Figure 9b), and add time and magnitude warping (see Figure 9c). We generate a signal-image pair such that the image is based on the signal data. We make use of the Gramian angular field (GAF) that transforms time-series into images. The time-series is defined as $\mathbf{x} = (x_1, \dots, x_n)$ for $n = 1,000$. The GAF creates a matrix of temporal correlations for each $(x_i, x_j)$ by rescaling the time-series in the range $[p, q]$ with $-1 \leq p < q \leq 1$ by

$$\hat{x}_i = p + (q - p) \cdot \frac{x_i - \min(\mathbf{x})}{\max(\mathbf{x}) - \min(\mathbf{x})}, \forall i \in \{1, \dots, n\}, \tag{4}$$

and computes the cosine of the sum of the angles for the Gramian angular summation field (GASF) Wang & Oates (2015) by

$$\text{GASF}_{i,j} = \cos\left(\phi_i + \phi_j\right), \forall i, j \in 1, \dots, n, \tag{5}$$

with $\phi_i = \arccos\left(\hat{x}_i\right), \forall i \in \{1, \dots, n\}$, being the polar coordinates. We generate image datasets based on signal data with different noise parameters ($b \in \{0.0, \dots, 1.95\}$) to show the influence of the image data on the classification accuracy. Figure 10 exemplarily shows the GASF plots for the noise parameters $b = [0, 0.5, 1.0, 1.5, 1.95]$. We present the GASF for the classes 0, 5 and 9 to show the dependency of the frequency of the signal data on the GASF.

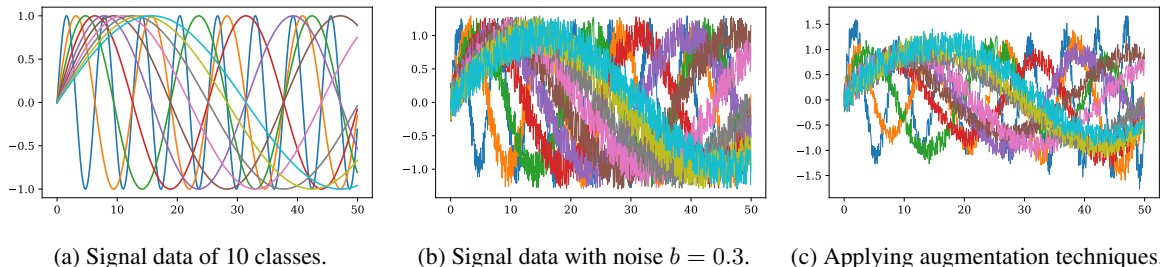

(a) Signal data of 10 classes.  (b) Signal data with noise $b = 0.3$.  (c) Applying augmentation techniques.

Figure 9: Plot of the 1D signal data for 10 classes.

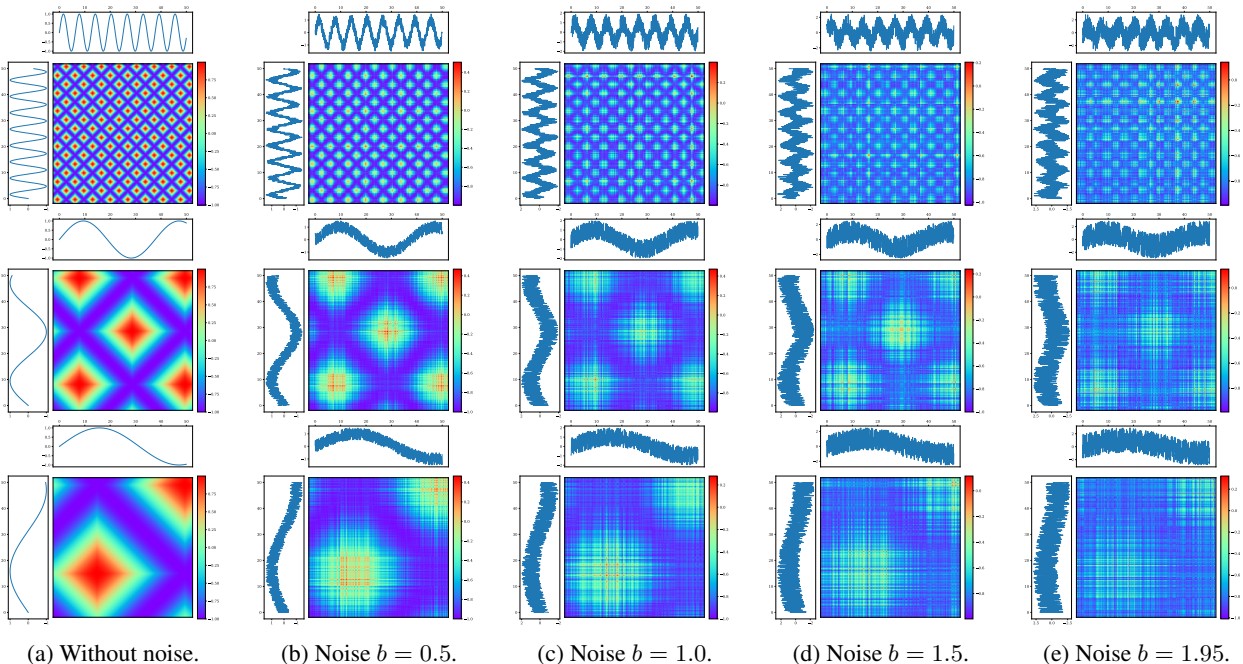

|                    |                    |                    |                    |                    |
| :----------------: | :----------------: | :----------------: | :----------------: | :----------------: |
| (a) Without noise. | (b) Noise $b = 0.5$. | (c) Noise $b = 1.0$. | (d) Noise $b = 1.5$. | (e) Noise $b = 1.95$. |

Figure 10: Plot of the Gramian angular summation field (GASF) based on 1D signal data with added noise for the classes 0 (top row), 5 (middle row) and 9 (botton row).

**Models.** We use the following models for classification. Our encoder for time-series classification consists of a 1D convolutional layer (filter size 50, kernel 4), a max pooling layer (pool size 4), batch normalization, and a dropout layer (20%). The image encoder consists of a layer normalization and 2D convolutional layer (filter size 200), and batch normalization with `ELU` activation. It follows a 1D convolutional layer (filter size 200, kernel 4), max pooling (pool size 2), batch normalization, and 20% dropout. For both models, it follows a common representation, i.e., an LSTM with 10 units, a Dense layer with 20 units, a batch normalization layer, and a Dense layer of 10 units (for 10 sinusoidal classes). These layers are shared between both models.

### A.3 Overview of Offline HWR Methods

In the following, we give a detailed overview of offline HWR methods to select a suitable lexicon and language model free method. There is no recent paper summarizing the work for offline HWR. For an overview of offline and online HWR datasets, see Plamondon & Srihari (2000); Hussain et al. (2015). Table 3 presents related work. Methods for offline HWR range from hidden markov models (HMMs) to deep learning techniques that became predominant such as convolutional neural networks (CNNs), temporal convolutional networks (TCNs) and recurrent neural networks (RNNs). RNN techniques are well explored including long short-term memorys (LSTMs), bidirectional LSTMs (BiL-STMs), and multidimensional RNNs (MDRNN, MDLSTM). Recent methods are generative adversarial networks (GANs) and Transformers. We note the use of a language model (LM) and its size $k$, and the data level the method works with, i.e., paragraph or full text level (P), line level (L) and word level (W). We present evaluation results for the IAM-OffDB Liwicki & Bunke (2005) and RIMES Grosicki & El-Abed (2011) datasets including the word error rate (WER) and character error rate (CER).

**HMMs.** Methods based on HMMs from last decades are Bertolami & Bunke (2018); Dreuw et al. (2011); Li et al. (2014); Pastor-Pellicer et al. (2015). Recently, España-Boquera et al. (2011) proposed `HMM+ANN`, a HMM modeled with Markov chains in combination with a multilayer perceptron (MLP) to estimate the emission probabilities. Kozielski et al. (2013) presented `Tandem GHMM` that uses moment-based image normalization, writer adaptation and discriminative feature extraction with an 3-gram open-vocabulary of size 50k with an LSTM for recognition. Doetsch et al. (2014) proposed an LSTM unit that controls the shape of the squashing function in gating units decoded in a hybrid HMM. This approach yields the best results based on HMMs.

Table 3: Evaluation results (WER and CER in %) of different methods on the IAM-OffDB Liwicki & Bunke (2005) and RIMES Grosicki & El-Abed (2011) datasets. We state information about the method and the size of the language model (LM). LN = layer normalization. P = paragraph or full text level. L = line level. W = word level. The table is sorted by year.

| | Method | Information | LM size $k$ | P | L | W | IAM-OffDB WER | CER | RIMES WER | CER |
|---|---|---|---|---|---|---|---|---|---|---|
| **HMM** | HMM+ANN España-Boquera et al. (2011) | Markov chain with MLP | w/ (5) | | | | 15.50 | 6.90 | - | - |
| | Tandem GHMM Kozielski et al. (2013) | GHMM and LSTM, writer adaptation | w/ (50) | | | × | 13.30 | 5.10 | 13.70 | 4.60 |
| | LSTM-HMM Doetsch et al. (2014) | Combination of LSTM with HMM | w/ (50) | | × | | 12.20 | 4.70 | 12.90 | 4.30 |
| **Multi-dim. LSTM** | 2DLSTM Graves & Schmidhuber (2008) | Combined MDLSTM with CTC | w/o | | | | 27.50 | 8.30 | 17.70 | 4.00 |
| | MDLSTM-RNN Bluche (2016) | 150 dpi | w/o | × | | | 29.50 | 10.10 | 13.60 | 3.20 |
| | | 150 dpi | w/ (50) | × | | | 16.60 | 6.50 | - | - |
| | | 300 dpi | w/o | × | | | 24.60 | 7.90 | 12.60 | 2.90 |
| | | 300 dpi | w/ (50) | × | | | 16.40 | 5.50 | - | - |
| | Voigtlaender et al. (2016) | GPU-based, diagonal MDLSTM | | | | | 9.30 | 3.50 | 9.60 | 2.80 |
| | SepMDLSTM Chen et al. (2017) | Multi-task approach | w/o | | | | 34.55 | 11.15 | 30.54 | 8.29 |
| | Bluche et al. (2017) | MDLSTM, attention | w/o | × | | | - | 16,20 | - | - |
| | | Line segmentation 150 dpi | w/o | | × | | - | 11.10 | - | - |
| | | Line segmentation 150 dpi | w/o | | × | | - | 7.50 | - | - |
| | MDLSTM Castro et al. (2018) | | | | | | 10.50 | 3.60 | - | - |
| **RNN** | BiLSTM Graves et al. (2009) | | w/ (20) | | | | 18.20 | 25.90 | - | - |
| | HMM+RNN Menasri et al. (2012) | Sliding win. Gaussian HMM, RNN | | | × | × | - | | 4.75 | - |
| | Dropout Pham et al. (2014) | LSTMs with dropout | w/o | | | | 35.10 | 10.80 | 28.50 | 6.80 |
| | Voigtlaender et al. (2015) | Maximum mutual information | | | | | 12.70 | 4.80 | 12.10 | 4.40 |
| | Bluche (2015) | | | | | | 10.90 | 4.40 | 11.20 | 3.50 |
| | | | w/ (50) | | | | 13.60 | 5.10 | 12.30 | 3.30 |
| | GCRNN Bluche & Messina (2017) | CNN+BiLSTM | w/ (50) | | | | 10.50 | 3.20 | 7.90 | 1.90 |
| | CNN-1DLSTM-CTC Puigcerver (2017) | CNN+BiLSTM+CTC (128 x W) | w/o | | × | | 18.40 | 5.80 | 9.60 | 2.30 |
| | | NN+BiLSTM+CTC | w/ (50) | | × | | 12.20 | 4.40 | 9.00 | 2.50 |
| | End2End Krishnan et al. (2018) | Without line level | w/ | | | | 16.19 | 6.34 | - | - |
| | | Line level | w/ | | × | | 32.89 | 9.78 | - | - |
| | SFR Wigington et al. (2018) | Text detection and segmentation | w/o | × | | | 23.20 | 6.40 | 9.30 | 2.10 |
| | CNN-RNN Dutta et al. (2018) | Unconstrained | w/o | | | | 12.61 | 4.88 | 7.04 | 2.32 |
| | | Full-Lexicon | w/ | | | | 4.80 | 2.52 | 1.86 | 0.65 |
| | | Text-Lexicon | w/ | | | | 4.07 | 2.17 | | |
| | | Unconstrained | w/o | | × | | 17.82 | 5.70 | 9.60 | 2.30 |
| | Chowdhury & Vig (2018) | Seq2seq, w/o LN | w/o | | | | 25.50 | 17.40 | 19.10 | 12.00 |
| | | w/ LN | w/o | | | | 22.90 | 13.10 | 15.80 | 9.70 |
| | | w/ LN + Focal Loss | w/o | | | | 21.10 | 11.40 | 13.50 | 7.30 |
| | | w/ LN + Focal Loss + Beam Search | w/o | | | | 16.70 | 8.10 | 9.60 | 3.50 |
| | Sueiras et al. (2018) | LSTM encoder-decoder, attention | | | | | 15.90 | 4.80 | - | - |
| | Chung & Delteil (2019) | ResNet+LSTM, segmentation | w/ | × | | | - | 8.50 | - | - |
| | Ingle et al. (2019) | BiLSTM | | | × | | 30.70 | 12.80 | - | - |
| | | GRCL | | | × | | 35.20 | 14.10 | - | - |
| | Michael et al. (2019) | Seq2seq CNN+BiLSTM (64 x W) | | | × | | - | 5.24 | - | - |
| | FPN Carbonell et al. (2019) | Feature Pyramid Network, 150 dpi | | | × | | - | 15.60 | - | - |
| | AFDM Bhunia et al. (2019) | AFD module | w/ | | | | 8.87 | 5.94 | 6.31 | 3.17 |
| **CNN** | Poznanski & Wolf (2016) | CNN + connected branches, CCA | w/ | | | | 6.45 | 3.44 | 3.90 | 1.90 |
| | GTR Yousef et al. (2018) | CNN+CTC (32 x W) | w/o | | × | | - | 4.90 | - | - |
| | OrigamiNet Yousef & Bishop (2020) | VGG (500x500) | × | × | | | - | 51.37 | - | - |
| | | VGG (500x500), w/o LN | w/o | × | × | | - | 34.55 | - | - |
| | | ResNet26 (500x500), w/o LN | w/o | × | × | | - | 10.03 | - | - |
| | | ResNet26 (500x500), w/ LN | w/o | × | × | | - | 7.24 | - | - |
| | | ResNet26 (500x500), w/o LN | w/o | × | × | | - | 8.93 | - | - |
| | | ResNet26 (500x500), w/ LN | w/o | × | × | | - | 6.37 | - | - |
| | | ResNet26 (500x500), w/o LN | w/o | × | × | | - | 76.90 | - | - |
| | | ResNet26 (500x500), w/ LN | w/o | × | × | | - | 6.13 | - | - |
| | | GTR-8 (500x500), w/o LN | w/o | × | × | | - | 72.40 | - | - |
| | | GTR-8 (500x500), w/ LN | w/o | × | × | | - | 5.64 | - | - |
| | | GTR-8 (750x750), w/ LN | w/o | × | × | | - | 5.50 | - | - |
| | | GTR-12 (750x750), w/ LN | w/o | × | × | | - | 4.70 | - | - |
| | DAN Wang et al. (2020) | Decoupled attention module | w/o | | × | | 19.60 | 6.40 | 8.90 | 2.70 |
| **GAN** | ScrabbleGAN Fogel et al. (2020) | Original data | w/o | | | | 25.10 | - | 12.29 | - |
| | | Augm. | w/o | | | | 24.73 | - | 12.24 | - |
| | | Augm + 100k synth. | w/o | | | | 23.98 | - | 11.68 | - |
| | | Augm + 100k synth. + Refine | w/o | | | | 23.61 | - | 11.32 | - |
| **Transformer** | Kang et al. (2020) | Self-attention for text/images | w/o | | × | | 15.45 | 4.67 | - | - |
| | FPHR Singh & Karayev (2021) | CNN encoder, Transformer decoder | w/o | × | | | - | 6.70 | - | - |
| | | With augmentation | w/o | × | | | - | 6.30 | - | - |
| **Other** | FST Messina & Kermorvant (2014) | Finite state transducer (lexicon) | n-gram | | | | 19.10 | - | 13.30 | - |

**RNNs: MDLSTMs.** The `2DLSTM` approach by Graves & Schmidhuber (2008) combines multidimensional LSTMs (MDLSTMs) with the CTC loss. The `MDLSTM-RNN` approach Bluche (2016) works at paragraph level by replacing the collapse layer by a recurrent version. A neural network performs implicit line segmentation by computing attention weights on the image representation. Voigtlaender et al. (2016) proposed an efficient GPU-based implementation of MDLSTMs by processing the input in a diagonal-wise fashion. `SepMDLSTM` Chen et al. (2017) is a multi-task learning method for script identification and HWR based on two classification modules by minimizing the CTC and negative log likelihood losses. While the MDLSTM by Bluche et al. (2017) contains covert and overt attention without prior segmentation, the Castro et al. (2018) integrated MDLSTMs within a hybrid HMM. However, these architectures come with quite an expensive computational cost. Furthermore, they extract features visually similar to those of convolutional layers Puigcerver (2017). `End2End` Krishnan et al. (2018) jointly learns text and image embeddings based on LSTMs.

**RNNs: LSTMs and BiLSTMs.** RNNs for HWR marked an important milestone reaching impressive recognition accuracies. Sequential architectures are perfect to fit text lines due to the probability distributions over sequences of characters and due to the inherent temporal aspect of text Kang et al. (2020). Graves et al. (2009) introduced the BiLSTM layer in combination with the CTC loss. Pham et al. (2014) showed that the performance of LSTMs can be greatly improved using `dropout`. Voigtlaender et al. (2015) investigated sequence-discriminative training of LSTMs using the maximum mutual information (MMI) criterion. While Bluche (2015) utilized a RNN with a HMM and a language model, Menasri et al. (2012) combined a RNN with a sliding window Gaussian HMM. `GCRNN` Bluche & Messina (2017) combines a convolutional encoder (aiming generic and multilingual features) and a BiLSTM decoder predicting character sequences. Also, Puigcerver (2017) proposed a CNN+BiLSTM architecture (`CNN-1DLSTM-CTC`) that uses the CTC loss. The start, follow, read (`SFR`) Wigington et al. (2018) model jointly learns text detection and segmentation. Dutta et al. (2018) used synthetic data for pre-training and image normalization for slant correction. The methods by Chowdhury & Vig (2018); Sueiras et al. (2018); Ingle et al. (2019); Michael et al. (2019) make also use of BiLSTMs. While Carbonell et al. (2019) uses a feature pyramid network (`FPN`), the adversarial feature deformation module (`AFDM`) Bhunia et al. (2019) learns ways to elastically warp extracted features in a scalable manner. Further methods that combine CNNs with RNNs are Liang et al. (2017); Sudholt & Fink (2018); Xiao & Cho (2016), while BiLSTMs are utilized in Carbune et al. (2020); Tian et al. (2019).

**TCNs.** TCNs uses dilated causal convolutions and have been applied to air-writing recognition by Bastas et al. (2020). As RNNs are slow to train, Sharma et al. (2020) presented a faster system which is based on text line images and TCNs with the CTC loss. This method achieves 9.6% CER on the IAM-OffDB dataset. Sharma & Jayagopi (2021) combined 2D convolutions with 1D dilated non-causal convolutions that offers a high parallelism with a smaller number of parameters. They analyzed re-scaling factors and data augmentation, and achieved comparable results for the IAM-OffDB and RIMES datasets.

**CNNs.** Poznanski & Wolf (2016) utilized a CNN with multiple fully connected branches to estimate its n-gram frequency profile (set of n-grams contained in the word). With canonical correlation analysis (CCA), the estimated profile can be matched to the true profiles of all words in a large dictionary. As most attention methods suffer from an alignment problem, Wang et al. (2020) proposed a decoupled attention network (`DAN`) that has a convolutional alignment module that decouples the alignment operation from using historical decoding results based on visual features. The gated text recognizer (`GTR`) Yousef et al. (2018) aims to automate the feature extraction from raw input signal with minimum required domain knowledge. The fully convolutional network without recurrent connections is trained with the CTC loss. Thus, the GTR module can handle arbitrary input sizes and can recognize strings with arbitrary length. This module has been used for `OrigamiNet` Yousef & Bishop (2020) that is a segmentation-free multi-line or full page recognition system. `OrigamiNet` yields state-of-the-art results on the IAM-OffDB dataset, and shows an improved performance of GTR over VGG and ResNet26. Hence, we use the GTR module as our visual feature encoder for offline HWR (see Section A.4).

**GANs.** Handwriting text generation (HTG) is a relatively new field. The first approach by Graves (2014) was a method to synthesize online data based on RNNs. The technique `HWGAN` by Ji & Chen (2020) extends this method by adding a discriminator $\mathcal{D}$. `DeepWriting` Aksan et al. (2016) is a GAN that is capable of disentangling style from content and thus making digital ink editable. Haines et al. (2016) proposed a method to generate handwriting based on a specific author with learned parameters for spacing, pressure and line thickness. Alonso et al. (2019) used a

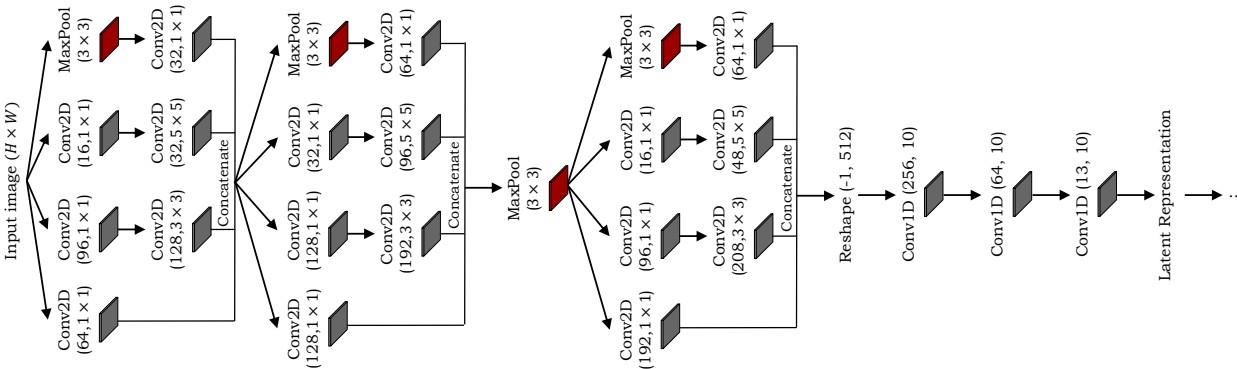

Figure 11: Offline HWR method based on Inception modules Szegedy et al. (2015).

BiLSTM to get an embedding of the word to be rendered, and added an auxiliary network as recognizer $\mathcal{R}$. The model is trained with a combination of an adversarial loss and the CTC loss. `ScrabbleGAN` by Fogel et al. (2020) is a semi-supervised approach that can generate arbitrarily many images of words with arbitrary length from a generator $\mathcal{G}$ to augment handwriting data and uses a discriminator $\mathcal{D}$ and recognizer $\mathcal{R}$. The paper proposes results for original data, with random affine augmentation, using synthetic images and refinement.

**Transformers.** RNNs prevent parallelization due to their sequential pipelines. Kang et al. (2020) introduced a non-recurrent model by the use of Transformer models by using multi-head self-attention layers at the textual and visual stages. Their method is unconstrained to any pre-defined vocabulary. For the feature encoder, they used modified ResNet50 models. The full page HTR (`FPHR`) method by Singh & Karayev (2021) uses a CNN as encoder and a Transformer as decoder with positional encoding.

## A.4  Details on Architectures for Offline HWR

In this section, we give details about the integration of `Inception` Szegedy et al. (2015), `ResNet` He et al. (2016) and `GTR` Yousef et al. (2018) modules into the offline HWR system. All three architectures are based on publicly available implementations, but we changed or adapted the first layer for the image input and the last layer for a proper input for our latent representation module.

**Inception.** Figure 11 gives an overview of the integration of the Inception module. The Inception module is part of the well known GoogLeNet architecture. The main idea is to consider how an optimal local sparse structure can be approximated by readily available dense components. As the merging of pooling layer outputs with convolutional layer outputs would lead to an inevitable increase in the number of output and would lead to computational blow up, we apply the Inception module with dimensionality reduction to our offline HWR approach Szegedy et al. (2015). The input image is of size $H \times W$. What follows is the Inception (3a), Inception (3b), a max pooling layer ($3 \times 3$) and Inception (4a). We add three 1D convolutional layers to get an output dimensionality of $400 \times 200$ as input for the latent representation.

**ResNet34.** Figure 12 gives an overview of the integration of the ResNet34 architecture. Instead of learning unreferenced functions, He et al. (2016) reformulated the layers as learning residual functions with reference to the layer inputs. This residual network is easier to optimize and can gain accuracy from considerably increased depth. The ResNet block let the layers fit a residual mapping denoted as $\mathcal{H}(\mathbf{x})$ with identity $\mathbf{x}$, and fits the mapping $\mathcal{F}(\mathbf{x}) := \mathcal{H}(\mathbf{x}) - \mathbf{x}$. The original mapping is recast into $\mathcal{F}(\mathbf{x}) + \mathbf{x}$. We reshape the output of ResNet34, add a 1D convolutional layer, and reshape the output for the latent representation.

**GTR.** Figure 13 gives an overview of the integration of the gated text recognizer (GTR) Yousef et al. (2018) module that is a fully convolutional network that uses batch normalization (BN) and layer normalization (LN) to regularize the training process and increase convergence speed. The module uses batch renormalization Ioffe (2017) on all BN layers. Depthwise separable convolutions reduce the number of parameters at the same/better classification performance.

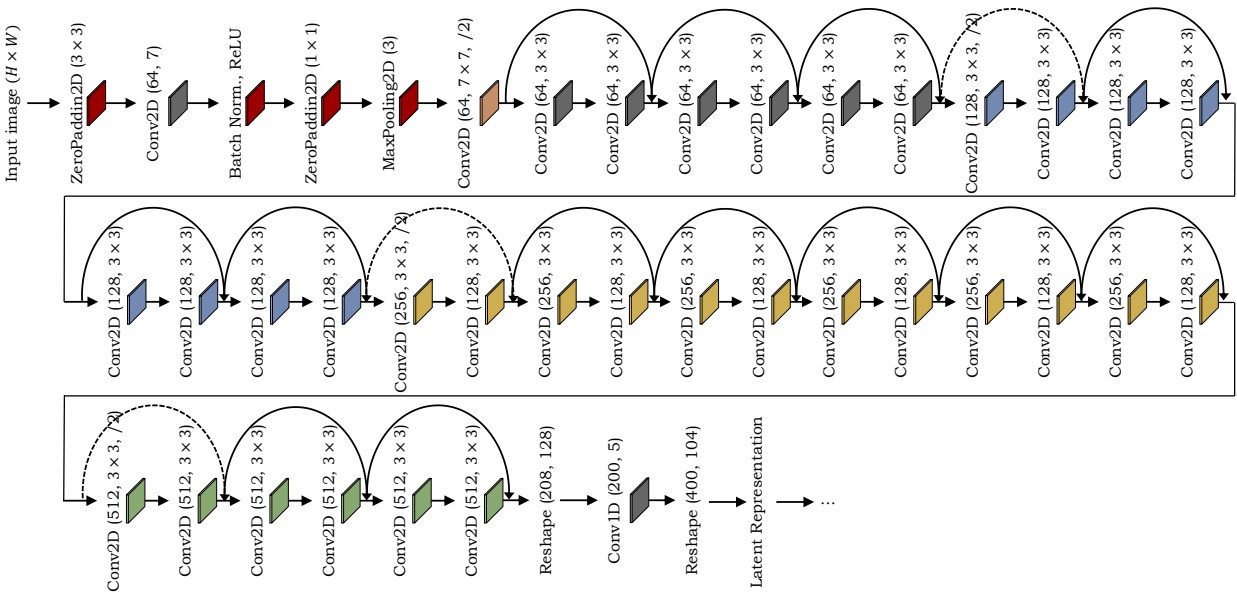

Figure 12: Offline HWR method based on the ResNet34 architecture He et al. (2016).

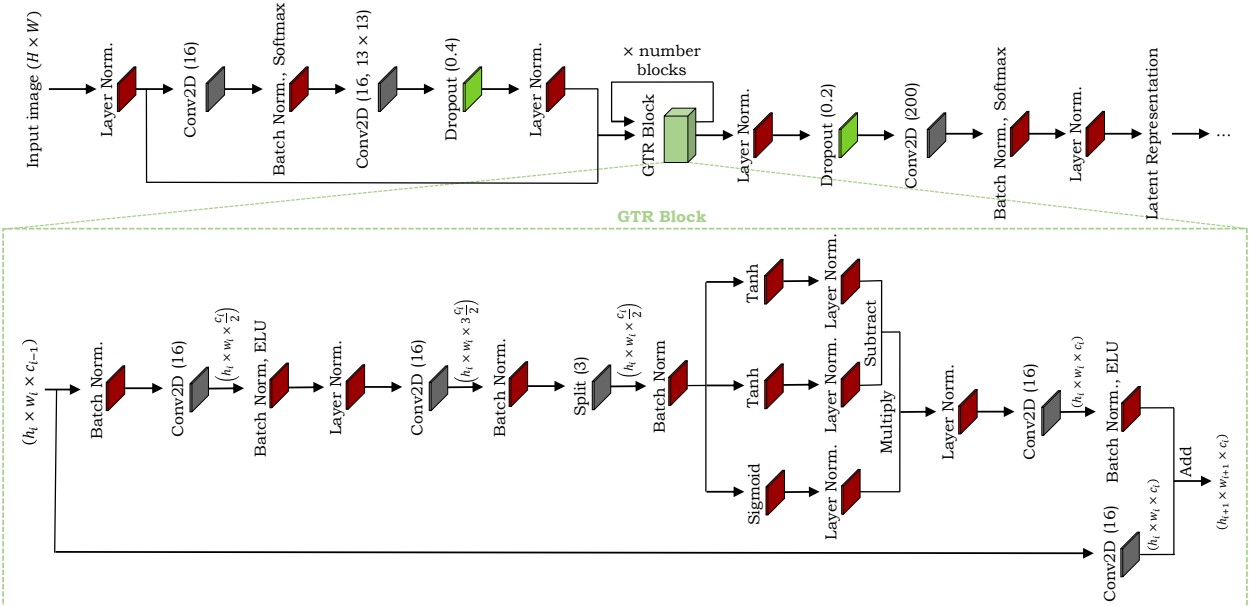

Figure 13: Offline HWR method based on the GTR architecture Yousef et al. (2018).

GTR uses spatial dropout instead of regular unstructured dropout for better regularization. After the input image of size $H \times W$ that is normalized follows a convolutional layer with Softmax normlization, a $13 \times 13$ filter, and dropout (40%). It follows a stack of 2, 4, 6 or 8 gate blocks that models the input sequence. Similar as Yousef et al. (2018), we add a dropout of 20% after the last GTR block. Lastly, we add a 2D convolutional layer of 200, a BN layer and a LN layer that is the input for our latent representation.

## A.5 Detailed HWR Evaluation

**Offline HWR Results.** Table 4 shows offline HWR results on our generated OffHW-German dataset and on the IAM-OffDB Liwicki & Bunke (2005) dataset. ScrabbleGAN Fogel et al. (2020) yields an WER of 23.61% on the

Table 4: Evaluation results (WER and CER in %) for the generated dataset with ScrabbleGAN Fogel et al. (2020) OffHW-German and the IAM-OffDB Liwicki & Bunke (2005) dataset. We propose writer-dependent (WD) and writer-independent (WI) results.

| | OffHW-German | | IAM-OffDB | |
|---|---|---|---|---|
| **Method** | **WER** | **CER** | **WER** | **CER** |
| ScrabbleGAN | - | - | 23.61 | - |
| Fogel et al. (2020) | | | | |
| OrigamiNet (12×GTR) | - | - | - | 4.70 |
| Yousef & Bishop (2020) | | | | |
| OrigamiNet (ours, 4×GTR) | 1.50 | 0.11 | 90.40 | 15.67 |
| Inception | 12.54 | 1.17 | - | - |
| ResNet | 13.05 | 1.24 | - | - |
| GTR (2 blocks), 1 conv. layer | 4.34 | 0.39 | - | - |
| GTR (2 blocks), 2 conv. layer | 5.02 | 0.44 | - | - |
| GTR (4 blocks), 1 conv. layer | 3.35 | 0.34 | 89.37 | 15.60 |
| GTR (4 blocks), 2 conv. layer | 2.52 | 0.24 | - | - |
| GTR (6 blocks) | 2.85 | 0.26 | - | - |
| GTR (8 blocks) | 4.22 | 0.38 | - | - |

Table 5: Evaluation results (WER and CER in %) for the generated OffHW-words500 and OffHW-wordsRandom datasets for one and two convolutional layers (c). We propose writer-dependent (WD) and writer-independent (WI) results.

| Method | OffHW-words500 | | | | OffHW-wordsRandom | | | |
|---|---|---|---|---|---|---|---|---|
| | WD | | WI | | WD | | WI | |
| (4×GTR) | **WER** | **CER** | **WER** | **CER** | **WER** | **CER** | **WER** | **CER** |
| $c = 1$ | 2.94 | 0.76 | 0.95 | 0.23 | 1.98 | 0.35 | 2.05 | 0.37 |
| $c = 2$ | 2.51 | 0.69 | 0.85 | 0.22 | 1.82 | 0.34 | 1.95 | 0.38 |

IAM dataset, while OrigamiNet Yousef & Bishop (2020) achieves an CER of 4.70% with 12 GTR modules. As the training takes more than one day for one epoch on the large OffHW-German dataset, we train OrigamiNet with four GTR modules, and achieve 0.11% CER on the generated dataset and 15.67% on the IAM dataset, which is higher than the model with 12 GTR modules. While the paper did not propose WER results, OrigamiNet yields only an WER of 90.40%. With our own implementation of four GTR modules and one convolutional layer for the common representation, our model achieves similar results. While GTR modules yield slightly lower CERs on the OffHW-German dataset than our architectures with Inception and ResNet modules, the WERs are significantly higher. Fine-tuning the architecture with four GTR modules and one ($c = 1$) or two ($c = 2$) convolutional layers on the OffHW-words500 and OffHW-wordsRandom datasets, yields better results for $c = 2$ than for $c = 1$ (see Table 5). While results for OffHW-wordsRandom are similar for writer-dependent (WD) and writer-independent (WI), WI results of the OffHW-words500 dataset are lower than WD results, as words with the same label appear in the training and test dataset. We use the weights of the fine-tuning as initial weights of the image model for the common representation learning.

**Online HWR Results.** Table 2 gives an overview of CRL results based on two convolutional layers ($c = 2$) for the common representation. Our CNN+BiLSTM contains three additional convolutional layers and outperforms the smaller CNN+BiLSTM by Ott et al. (2022) on the WD classification tasks. Without triplet loss, $\mathcal{L}_{PC}$ yields the best results on the OnHW-wordsRandom dataset. The triplet loss partly marginally decreases results and partly improves results on the OnHW-words500 dataset. In conclusion, two convolutional layers for the common representation decreases results, while here the triplet loss does not have a positive impact.

Table 6: Evaluation results (WER and CER in %, averaged over five splits) of the baseline MTS-only technique and our cross-modal techniques for the inertial-based OnHW datasets Ott et al. (2022) with and without mutated vowels (MV) for two convolutional layers $c = 2$. We propose writer-(in)dependent (WD/WI) results.

| | OnHW-words500 | | | | OnHW-wordsRandom | | | |
| | WD | | WI | | WD | | WI | |
| Method | WER | CER | WER | CER | WER | CER | WER | CER |
|---|---|---|---|---|---|---|---|---|
| Small CNN+BiLSTM, $\mathcal{L}_{CTC}$, w/ MV | 51.95 | 17.16 | 60.91 | 27.80 | 41.27 | 7.87 | 84.52 | 35.22 |
| CNN+BiLSTM (ours), $\mathcal{L}_{CTC}$, w/ MV | 42.81 | 13.04 | 60.47 | 28.30 | 37.13 | 6.75 | 83.28 | 35.90 |
| CNN+BiLSTM (ours), $\mathcal{L}_{CTC}$, w/o MV | 42.77 | 13.44 | 59.82 | 28.54 | 41.52 | 7.81 | 83.54 | 36.51 |
| $\mathcal{L}_{MSE}$ | 39.79 | 12.14 | 60.35 | 28.48 | 39.98 | 7.79 | 83.50 | 36.92 |
| $\mathcal{L}_{CS}$ | 43.40 | 13.70 | 59.31 | 27.99 | 40.31 | 7.68 | 83.68 | 36.30 |
| $\mathcal{L}_{PC}$ | 38.90 | 11.60 | 60.77 | 28.45 | **39.93** | **7.60** | **83.19** | **35.83** |
| $\mathcal{L}_{KL}$ | **37.25** | **11.29** | 65.10 | 31.26 | 41.81 | 8.22 | 84.40 | 38.93 |
| $\mathcal{L}_{trpl,2}(\mathcal{L}_{MSE})$ | 41.16 | 12.71 | 58.65 | 28.19 | 41.16 | 8.03 | 85.38 | 39.49 |
| $\mathcal{L}_{trpl,2}(\mathcal{L}_{CS})$ | 42.74 | 13.43 | **58.13** | **27.62** | 41.49 | 8.18 | 85.24 | 38.75 |
| $\mathcal{L}_{trpl,2}(\mathcal{L}_{PC})$ | 39.94 | 12.19 | 62.76 | 30.68 | 41.58 | 8.18 | 85.18 | 38.53 |
| $\mathcal{L}_{trpl,2}(\mathcal{L}_{KL})$ | 38.34 | 11.77 | 67.08 | 33.84 | 41.87 | 8.33 | 86.34 | 40.37 |

Table 7: Evaluation results (WER and CER in %, averaged over five splits) of the baseline MTS-only technique and our cross-modal techniques for the inertial-based OnHW datasets Ott et al. (2022) with and without mutated vowels (MV) for two convolutional layers $c = 1$. We propose writer-(in)dependent (WD/WI) results.

| | OnHW-words500-L | | | | OnHW-wordsRandom-L | | | |
| | WD | | WI | | WD | | WI | |
| Method | WER | CER | WER | CER | WER | CER | WER | CER |
|---|---|---|---|---|---|---|---|---|
| InceptionTime, $\mathcal{L}_{CTC}$, w/ MV | 49.70 | 14.02 | 100.00 | 96.06 | 48.10 | 8.63 | 100.00 | 95.93 |
| CNN+BiLSTM, $\mathcal{L}_{CTC}$, w/ MV | 14.20 | 3.30 | 94.40 | 71.41 | 30.20 | 4.86 | 100.00 | 83.51 |
| CNN+BiLSTM, $\mathcal{L}_{CTC}$, w/o MV | 12.94 | 3.33 | 89.07 | 62.07 | 30.89 | 5.26 | 100.00 | 81.15 |
| $\mathcal{L}_{MSE}$ | **11.62** | 2.77 | 90.65 | 67.90 | 30.53 | 4.93 | 100.00 | 81.99 |
| $\mathcal{L}_{CS}$ | 14.92 | 3.53 | 94.14 | 65.10 | 29.06 | 4.87 | 100.00 | 83.94 |
| $\mathcal{L}_{PC}$ | 12.29 | 3.04 | 91.33 | 60.89 | 27.32 | **4.47** | 100.00 | 85.09 |
| $\mathcal{L}_{KL}$ | 11.37 | 2.57 | 93.02 | 66.64 | 29.61 | 4.91 | 100.00 | 81.28 |
| $\mathcal{L}_{trpl,2}(\mathcal{L}_{MSE})$ | 11.97 | 2.83 | **84.34** | **57.84** | **27.19** | 4.79 | **99.87** | 82.60 |
| $\mathcal{L}_{trpl,2}(\mathcal{L}_{CS})$ | 11.65 | **2.63** | 94.70 | 67.69 | 28.39 | 4.62 | 100.00 | 83.44 |
| $\mathcal{L}_{trpl,2}(\mathcal{L}_{PC})$ | 13.02 | 2.94 | 89.86 | 60.26 | 30.22 | 4.81 | 100.00 | 84.29 |
| $\mathcal{L}_{trpl,2}(\mathcal{L}_{KL})$ | 13.55 | 3.22 | 97.86 | 76.54 | 28.14 | 4.71 | 100.00 | **80.81** |

**Transfer Learning on Left-Handed Writers.** To adapt the model to left-handed writers that are typically under-represented in the real-world, we make use of the left-handed datasets OnHW-words500-L and OnHW-wordsRandom-L proposed by Ott et al. (2022). These datasets contain recordings of two writers with 1,000 samples, respectively 996. As baseline we pre-train the MTS-only model on the right-handed datasets and post-train the left-handed datasets for 500 epochs (see the first two rows of Table 7). As these datsets are rather small, the models can overfit on these specific writers and achieve a very low CER of 3.33% on the OnHW-words500-L datasets and 5.26% CER on the OnHW-wordsRandom-L dataset without mutated vowels (MV) for the writer-dependent (WD) tasks, but the models can not generalize on the writer-independent (WI) tasks: 62.07% CER on the OnHW-words500-L dataset and 81.15% CER on the OnHW-wordsRandom-L dataset. Hence, we focus on the WD tasks. For comparison, we use the state-of-the-art time-series classification technique InceptionTime Fawaz et al. (2019) with $depth = 11$ and $nf = 96$ (without pre-training). Our CNN+BiLSTM clearly outperforms InceptionTime. We use the weights of the pre-training with the offline handwriting datasets and again post-train on the left-handed datasets with $c = 1$. Using the weights of the cross-modal learning without the triplet loss, can decrease the error rates up to 2.57% CER with $\mathcal{L}_{KL}$, respectively 4.47% CER with $\mathcal{L}_{PC}$. Using the triplet loss $\mathcal{L}_{trpl,2}(\mathcal{L}_{CS})$, we can further decrease the CER to 2.63% for the OnHW-words500-L dataset. In conclusion, due to the use of the weights of the cross-modal setup, the model can adapt faster to new writers and generalize better to unseen words due to the triplet loss.

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
