# OpenReview forum: "Auxiliary Cross-Modal Common Representation Learning with Triplet Loss Functions for Online Handwriting Recognition"
_TMLR — Rejected by TMLR_

### Review · Reviewer_cbhN · 2022-05-25

**Summary Of Contributions:**

This paper proposes an online handwriting recognition model by using the concept of common representation learning (CRL). Especially, this paper uses the triplet loss between two modalities, images and multivariate time-series (MTS) captured by  IMU censors, between two modality-specific encoders. The proposed model consists of two modules: the offline auxiliary network  (image encoder) and the main network (IMU encoder). The auxiliary network uses a pre-trained ScrabbleGAN, that generates handwriting images, as a data generator. Using the data generated by ScrabbleGAN, the auxiliary network is trained by the CTC word classification loss. The main network is trained by IMU data points and the CTC word prediction loss. This paper proposes to use the triplet loss between the image encoder and the IMU encoder for making a shared embedding space between two modalities. The experimental results show that using the triplet loss helps the final performances.

**Broader Impact Concerns:**

I don't have any ethical concerns regarding this paper. The broader impact section looks reasonable to me.

**Requested Changes:**

I think this paper needs heavy paper revisions and would like to suggest to submit as a new submission.

Especially, this paper needs to improve its writing quality and presentation, especially for the position of this paper.
In my opinion, this paper would be better to be positioned as a HWR method. In this case, as my review in the weakness section, this paper needs heavy revisions. Please read my comment in `Quality of the writing`.

The lack of novelty of the proposed method is the most strong weakness.
I cannot find any reason why the proposed method is specifically a good method and how to solve the proposed HWR tasks. Compared to a similar field, OCR, I think

In terms of the evaluation, I don't think the experimental results support the proposed method well. It needs more comparisons with other baselines. Because I am a novice of HCR, it is hard to denote what baselines are specifically required, but in my opinion, other recent HCR methods such as "Full Page Handwriting Recognition via Image to Sequence Extraction" (ICDAR 2021), "Fast Multi-language LSTM-based Online Handwriting Recognition" IJDAR should be compared as well.

Even if the proposed method shows better scores than previous methods, the proper analysis should be required. The triplet loss is actually the key of the performance improvements? or the choice of sub-modules is the key of the performance improvements? (e.g., data augmentation by GAN).

Overall, I recommend to revise the paper and resubmit as "new".


**[Minor changes]**

Please use `\citep`. The citation format confuses me a lot during reading the paper. For example:

> the domain with cross-modal learning allows to use information in all domains Ranjan et al. (2015).

It should be:

> the domain with cross-modal learning allows to use information in all domains (Ranjan et al. 2015).


**Strengths And Weaknesses:**

## [Strength]

- The paper shows better results on OnHW datasets compared to the baseline method (Ott et al. 2022c).

## [Weakness]

### [Novelty: There already exists a lot of triplet-based cross-modal learning methods without considering the other modality during inference]

In terms of the general machine learning community, the proposed method is not specifically novel compared to the previous multi-modal learning methods. While the paper argues that `A limitation of CRL is that most approaches require the availability of all modalities at inference time` and `While CRL learns representations from all modalities, single-modal learning commonly uses pair-wise learning`, they are not true.

For example, in image-to-caption cross-modal retrieval tasks (mapping images and texts in the shared embedding space => often called visual semantic embedding, VSE), the most common design choice is the separated encoders that allows the separated inference without the other modality (e.g., the image encoder can encode images without texts and vice versa). Moreover, the triplet loss is already standard in VSE methods [A]. Please refer the most recent VSE methods, such as [B].

[A] VSE++: Improving Visual-Semantic Embeddings with Hard Negatives , F. Faghri, D. J. Fleet, J. R. Kiros, S. Fidler, Proceedings of the British Machine Vision Conference (BMVC), 2018.

[B] Chen, Jiacheng, et al. "Learning the best pooling strategy for visual semantic embedding." Proceedings of the IEEE/CVF Conference on Computer Vision and Pattern Recognition. 2021.

The image-to-caption cross-modal retrieval (CMR) is widely studied for recent years. I cannot list up the whole list of the papers, but I would like to say that CMR is very active area, and they do not suffer from the main motivation of this paper. Therefore, if this paper would like to have a position of "common representation learning", at least it should contain a lot of VSE methods, such as:

- Andrea Frome, Greg S Corrado, Jon Shlens, Samy Bengio, Jeff Dean, Marc’Aurelio Ranzato, and Tomas Mikolov. Devise: A deep visual-semantic embedding model. In Proc. NeurIPS, pages 2121–2129, 2013.
- Peter Young, Alice Lai, Micah Hodosh, and Julia Hockenmaier. From image descriptions to visual denotations: New similarity metrics for semantic inference over event descriptions. ACL, 2:67–78, 2014.
- Ryan Kiros, Ruslan Salakhutdinov, and Richard S Zemel. Unifying visual-semantic embeddings with multimodal neural language models. arXiv preprint arXiv:1411.2539, 2014.
- Fartash Faghri, David J Fleet, Jamie Ryan Kiros, and Sanja Fidler. VSE++: Improving visual-semantic embeddings with hard negatives. In Proc. BMVC, 2018.
- Jiuxiang Gu, Jianfei Cai, Shafiq R Joty, Li Niu, and Gang Wang. Look, imagine and match: Improving textual-visual cross-modal retrieval with generative models. In Proceedings of the IEEE conference on computer vision and pattern recognition, pages 7181–7189, 2018.
- Kuang-Huei Lee, Xi Chen, Gang Hua, Houdong Hu, and Xiaodong He. Stacked cross attention for image-text matching. In Proc. ECCV, 2018.
- Yan Huang, Qi Wu, Chunfeng Song, and Liang Wang. Learning semantic concepts and order for image and sentence matching. In Proceedings of the IEEE Conference on Computer Vision and Pattern Recognition, pages 6163–6171, 2018.
- Kunpeng Li, Yulun Zhang, Kai Li, Yuanyuan Li, and Yun Fu. Visual semantic reasoning for image-text matching. In Proc. ICCV, pages 4654–4662, 2019.
- Yale Song and Mohammad Soleymani. Polysemous visual-semantic embedding for cross-modal retrieval. In Proc. CVPR, pages 1979–1988, 2019.
- Jonatas Wehrmann, Douglas M Souza, Mauricio A Lopes, and Rodrigo C Barros. Language-agnostic visual-semantic embeddings. In Proceedings of the IEEE/CVF International Conference on Computer Vision, pages 5804–5813, 2019.
- Hao Wu, Jiayuan Mao, Yufeng Zhang, Yuning Jiang, Lei Li, Weiwei Sun, and Wei-Ying Ma. Unified visual-semantic embeddings: Bridging vision and language with structured meaning representations. In Proceedings of the IEEE/CVF Conference on Computer Vision and Pattern Recognition
- Haoran Wang, Ying Zhang, Zhong Ji, Yanwei Pang, and Lin Ma. Consensus-aware visual-semantic embedding for image-text matching. In Proc. ECCV, 2020.
- Tianlang Chen, Jiajun Deng, and Jiebo Luo. Adaptive offline quintuplet loss for image-text matching. In Proc. ECCV, 2020.
- Haiwen Diao, Ying Zhang, Lin Ma, and Huchuan Lu. Similarity reasoning and filtration for image-text matching. In Proc. AAAI, 2021.
- Sanghyuk Chun, Seong Joon Oh, Rafael Sampaio De Rezende, Yannis Kalantidis, and Diane Larlus. Probabilistic embeddings for cross-modal retrieval. In Proc. CVPR, 2021.
- Jiacheng Chen, Hexiang Hu, Hao Wu, Yuning Jiang, and Changhu Wang. Learning the best pooling strategy for visual semantic embedding. In Proc. CVPR, 2021.
- Zhenyu Huang, Guocheng Niu, Xiao Liu, Wenbiao Ding, Xinyan Xiao, hua wu, and Xi Peng. Learning with noisy correspondence for cross-modal matching. In A. Beygelzimer, Y. Dauphin, P. Liang, and J. Wortman Vaughan, editors, Proc. NeurIPS, 2021.
- Ali Furkan Biten, Andres Mafla, Lluís Gómez, and Dimosthenis Karatzas. Is an image worth five sentences? a new look into semantics for image-text matching. In Proceedings of the IEEE/CVF Winter Conference on Applications of Computer Vision, pages 1391–1400, 2022
- Alec Radford, Jong Wook Kim, Chris Hallacy, Aditya Ramesh, Gabriel Goh, Sandhini Agarwal, Girish Sastry, Amanda Askell, Pamela Mishkin, Jack Clark, Gretchen Krueger, and Ilya Sutskever. Learning transferable visual models from natural language supervision. In Marina Meila and Tong Zhang, editors, Proc. ICML, volume 139 of Proceedings of Machine Learning Research


Furthermore, when I checked the previous benchmark paper (Ott et al. 2022c), the proposed method does not specifically outperform the other methods. For example, `InceptionTime + BiLSTM` model in Table 4 of Ott et al. (2022c) shows comparable or even outperformed performance compared to the proposed method (e.g., the best proposed model shows 6.98 OnHW-wordsRandom WD CER and the inception model shows 6.39 in the same setting). Therefore, I think it is difficult to argue that the proposed method shows specifically outperformed performances compared to the existing methods.


### [Quality of the writing]

I think the quality of the writing has a lot of rooms for the improvements.

**Introduction**

- If this paper would like to be positioned as a machine learning paper, as my comments in `Novelty`, the main arguments need more literature survey, especially on cross-modal retrieval for image-text matching.
- If this paper would like to be placed into online handwriting recognition (OHR), it needs a lot of contents related to OHR. What is OHR? What is the common approaches by the previous methods? What is the problem / limitation of the previous methods?

I feel that this paper does not satisfy any of the aboves.

**Related works**

- This paper omits many related works in CMR. Please read my comments in the above
- I don't think "deep metric learning" subsection is required here. If it is really important subsection, it should contain more recent published papers. Check 'Musgrave, Kevin, Serge Belongie, and Ser-Nam Lim. "A metric learning reality check." European Conference on Computer Vision. Springer, Cham, 2020.' for the list of recent metric learning methods.
- "The classical cross- entropy (CE) loss, however, is not useful for DML as it ignores how close each point is to its class centroid (or how far apart from other class centroids)" it is wrong. CE is the most popular choice in face recognition tasks with metric learning. Please check "Liu, Weiyang, et al. "Sphereface: Deep hypersphere embedding for face recognition." Proceedings of the IEEE conference on computer vision and pattern recognition. 2017." for more details.
- I think this paper needs a subsection for the comparison of HWR methods, and the difference between them and the proposed method.

**Method**

- They are not actually "Method". The actual method is described in 4.2 (Experiments)
- I would like to encourage to move the details of the proposed method in 4.2 into Sec 3.
- I think the details of the sub-modules, such as ScrabbleGAN and GTR, are required. I cannot get any detailed information of them, despite of their importance in the method.
- The meaning of `Softmax attention` is unclear. It would be `softmax` IMO.

**Experiments**

- This section acutally does not describe the experiments, but the models
- This section should explain the details of the target task (including datasets and evaluation metrics) and comparison methods
- The details of the benchmark datasets look not enough to me. There could be readers who are not familiar with HCR tasks (like me).

**Experimental results**

- Figure 7 could be a table, instead of line chart
- I really cannot get any information from Table 1. Why the feature map figure is required?


### Comparison methods

I am a novice in HCR, but even with my quick google search, I can find a number of papers for solving HCR. For example

- "Full Page Handwriting Recognition via Image to Sequence Extraction" (ICDAR 2021)
- "Fast Multi-language LSTM-based Online Handwriting Recognition" IJDAR

I think the proposed method should be compared with more HCR methods to support the superiority of the method. However, I cannot find any discussion for related works or any experiment for the comparions.


### Limitation of the method

As denoted in the paper, the proposed method relys a lot on ScrabbleGAN.

> We compare to results without MV as ScrabbleGAN is pretrained on IAM-OffDB that does not contain MV, and hence, such words cannot be generated.

In other words, the method is only applicable when ScrabbleGAN is trainable. It limits the possibility of the method.

---

> ### Author Response · Authors · 2022-06-03
> **Answer to feedback from reviewer cbhN**
>
> We thank the reviewer for taking the time to point out options to improve our manuscript and for the detailed feedback.
>
> (W1) We will clarify the limitation of CRL and the availability of modalities at inference time by referring to citations [A, B] and show the differences to these approaches in our related work section.
>
> (W2) Indeed, the field of common representation is widely studied and there exist a lot of references of triplet learning. We cited the - in our eyes - most important references for our application, but understand that references are missing in the field of cross-modal learning. We will enhance our manuscript by adding more references for the field of visual semantic embedding and present differences to the works suggested by reviewer cbhN.
>
> (W3) Reviewer cbhN addresses the two options to position the paper as a machine learning or handwriting recognition (HWR) paper. As the contributions of this paper are very HWR-related due to the text-specific triplet selection and as there already exist much research on cross-modal learning with triplet learning, we suggest placing our manuscript as an HWR paper with focus on cross-modal learning. For this, we will add further discussions on state-of-the-art methods of HWR, i.e., offline and online HWR. Currently, an overview of related work is included in the Appendix A.3. We will also further suggest adding more experiments on state-of-the-art HWR techniques such as InceptionTime+BiLSTM to show the improvements of our method. We thank the reviewer for pointing this out.
>
> (W4) We will restructure our method and experiments sections by moving the actual method from Section 4.2 into Section 3.
>
> (W5) In the current version of the manuscript, there are only the most important details of ScrabbleGAN included. We will add further details from ScrabbleGAN to make it better understandable.
>
> (W6) Appendix A.3 summarizes related work of offline HWR and compares state-of-the-art results on the IAM-OffDB and RIMES datasets. Here, we showed that GTR from OrigamiNet (Yousef & Bishop, 2020) achieves very good results without a language model. We would like to point out that in the Appendix A.4, we already proposed more details on the GTR architecture.
>
> (W7) We will replace the phrase “Softmax attention” with “softmax”.
>
> (W8) We will provide more details on the target task and benchmark datasets in the experiments section.
>
> (W9) We will add a table of the experimental results for the synthetic dataset to make the results easier comparable.
>
> (W10) The suggested papers for HWR are currently included in the appendix. The paper "Full Page Handwriting Recognition via Image to Sequence Extraction" is not applicable for online HWR from sensor-enhanced pens as it classifies images of pages. The paper "Fast Multi-language LSTM-based Online Handwriting Recognition" solves the online HWR task for the typical trajectory-based classification and not the classification task of signal data from sensor-enhanced pens. Hence, Ott et al. (2020c) benchmarked different time-series classification techniques. We will further elaborate to include state-of-the-art HWR techniques.
>
> (W11) We would like to thank the reviewer for pointing out this possible limitation. The limitation of the method is the requirement of an image-based dataset in the same language. As the OnHW-words and OnHW-wordsRandom datasets are written in german and contain word labels with mutated vowels, a similar image-based german dataset is required that does not exist currently. Closest to the OnHW dataset is the IAM-OffDB dataset that does not contain mutated vowels, and hence, the OCR method cannot be pre-trained on words with mutated vowels. In conclusion, the method is not limited by ScrabbleGAN, but by the image-based dataset required for pre-training. The GTR method could also be directly pre-trained on the IAM-OffDB dataset, but we assume less generalized results than for our generated dataset.
>
> (W12) We will use \citep as citation style for better readability.

---

### Review · Reviewer_6y6a · 2022-06-03

**Summary Of Contributions:**

The authors present an approach for handwritten digit recognition (HWR). They are mostly interested in online HWR, and specifically the case where a sensor-enhanced pen is used, while writing on normal paper (instead of a stylus pen on a touch screen surface).
They propose to learn a joint space and representation between pairs of image and time-series data, ie
offline HWR from generated images (OCR) and online HWR from sensor-enhanced pens, by learning a common representation between both modalities. They learn using a triplet loss.

**Broader Impact Concerns:**

Nothing concerning, the authors discuss data recording practices in their statement.

**Requested Changes:**

1) Justification/reduction of novel acronyms and improper use of terms. Can the authors justify why the use yet-another term/acronym for joinltly learning a space from two modalities? "Common representation learning (CRL)" is not a commonly used term and will eventually make it harder for people to find this paper. "Cross-modal" captures that, ie "cross-modal representation learning" (used as a title for section 2.1) seems like a more concise term to me. This is also the case for the sentence "DML is an interesting approach for continual learning" on page1; continual learning is a term that is widely used for a different set of tasks.

2) Notation. The notation is also highly non-standard and there fore at times hard to follow (although not wrong). But using commonly used variable names. eg: y for class label (instead of v, taht is easily confused with te variable u used for timeseries),  h/w  for image dimensions (instad of the arbitrary o/p). The feature space is defined as \real^{q,w} - it is unclear what these dimensions are. p is both a dimensionality for images and also denotes "positive" in Eq1/2

3) The abstract can be highly improved. Most of it is spent on defining the very common task of learning a joint cross-modal space and the triplet loss. In my opinion it could focus on the peculiarities of the specific application that the paper focuses on.

4) Fig 4 is very unclear to me - a "representation" is a feature vector not a model (the labels are referring to models now). I assume that the authors mean that the "representation" is the output of the corresponding box - it can be clarified.

A general note:
This paper adopts a baseline method for cross-modal learning and applies it to a specific application and specific set of modalities. It is therefore important to rephrase/rewrite the paper as such, and focus on the peculiarities of the application and the training data.
Right now, the method section 3.1 is basically notation and a list of common DML losses, while section 3.2 presents a triplet loss across the two modalities. It could instead focus on the peculiarities of learning with time-series, where some novelty of the paper could be found.
Moreover, as mentioned above, the paper would be far stronger if some more recent DML losses and approaches were adopted or tested, like mixing, contrastive losses or proxy-based losses[E,F,G]

**Strengths And Weaknesses:**

**Strengths:**

S1) to my knowledge it is an interesting idea to learn a joint space between time-series data from sensor-enhanced pen and text images as it could enable learning a stronger representation for both online and offline HWR.

S2) the use of GANs to create time-series+image pair data for Offline HWR is an interesting idea

**Weaknesses:**

The main criteria for acceptance at TMLR are for the claims of the paper to be "supported by accurate, convincing and clear evidence". In general there are a lot of unsupported claims in the paper.

W1) "the first to propose the triplet loss for sequence-based classification (i.e., words)." this is not really accurate  - The triplet loss has been extensively used before for cross-modal learning, see e.g. [A, B, C] below (and there are many more) while even sequence based tasks like ASR have used triplet losses. [D, or Zeng et al. (2020)] Even if it wasnt exactly used for these two specific modalities, It is a stretch to present it this way.

W2) the fact that a joint representation is learned is not fully exploited: they authors focus on improving time-series classification via this common space; it is however also important to understand how this space would also work for OCR - Given that a lot of data exist of OCR, wouldn't the common space further improve if one starts from a state-of-the-art OCR model, ie pretrains the image encoder and fine-tunes it with the joint data?

W3) The authors transform time-series to images via GASF and process them with a CNN this brings the application very close to others, eg DeepTripletNN Zeng et al. (2020) -  the authors should clearly discuss the differences and maybe even compare wherever possible.

W4) Section 4.1 is really not justified. IS this a sanity check? I cannot understand why and how the synthetic signal data from Fig 3a correspond to something related to HWR.

W4) the experimental evaluation is lacking - there is only a comparison table for different basic DML loss functions and no comparisons to other related works. I understand that maybe no other paper solves this specific task, but given how basic the prposed technical approach is from a learning perpsective, the authors can make some effort to compare to other recent methods for cross-modal represenation learning. E.g. given the data/pairs and the time-series-to-image encoder that the authors use, other methods can be used on top beyond a triplet loss and the other basic losses in Table 2. eg like mixing, contrastive losses or proxy-based losses [E, F, G]

**References**


[A] Gordo, Albert, and Diane Larlus. "Beyond instance-level image retrieval: Leveraging captions to learn a global visual representation for semantic retrieval." Proceedings of the IEEE conference on computer vision and pattern recognition. 2017.

[B]  Deng, Cheng, et al. "Triplet-based deep hashing network for cross-modal retrieval." IEEE Transactions on Image Processing 27.8 (2018): 3893-3903.

[C] Zhang, Ji, et al. "Large-scale visual relationship understanding." Proceedings of the AAAI conference on artificial intelligence. Vol. 33. No. 01. 2019.

[D] Bredin, Hervé. "Tristounet: triplet loss for speaker turn embedding." 2017 IEEE international conference on acoustics, speech and signal processing (ICASSP). IEEE, 2017.

[E] Kim, Sungyeon, et al. "Proxy anchor loss for deep metric learning." Proceedings of the IEEE/CVF Conference on Computer Vision and Pattern Recognition. 2020.

[F] Venkataramanan, Shashanka, et al. "AlignMix: Improving representation by interpolating aligned features." Proceedings of the IEEE/CVF Conference on Computer Vision and Pattern Recognition. 2022.

[G] Zhang, Haozhi, et al. "Rethinking Deep Contrastive Learning with Embedding Memory." arXiv preprint arXiv:2103.14003 (2021).

---

> ### Author Response · Authors · 2022-06-03
> **Answer to feedback from reviewer 6y6a**
>
> We thank the reviewer for the detailed feedback that we address in the following comments.
>
> (W1) We understand that the formulation is misleading. We will add reference [A, B, C] from reviewer 6y6a and present a clear separation between the use of the triplet loss from references [A, B, C, (Zeng et al. (2020)] to our method. We will add this discussion in the related work section.
>
> (W2) We tempted to clearly show that a joint representation is learned between both modalities by presenting commonalities and differences in the feature embeddings of pairs with an edit distance of 0 up to 3 shown in Table 1. Furthermore, we visualized the common feature embeddings of the image and time-series data with t-SNE in Figure 8b. Here, it is visible that both modalities are structured (i.e., green and orange embeddings of image and time-series data alternate). How does the reviewer suggest to present the common representation in a clear way? We would assume that this common representation would also work for the OCR task by exploiting time-series data. As the classification error of the OCR task is already very low due to a high amount of generated training data, we assume that the influence of time-series data is small. The influence of the common representation with time-series data would be increased by a small OCR dataset together with a large time-series dataset.
>
> (W3) As stated previously, we will discuss in more detail the differences to the DeepTripletNN approach by Zeng et al. (2020).
>
> (W4) Section 4.1 is a sanity check and is not related to HWR. The goal is to present the influence of the common representation between two modalities for an increasing amount of noise (i.e., the same information of the class label is present in both modalities). We showed that by exploiting the image data, the time-series model converges faster with higher classification accuracy. The classification accuracy improves for lower introduced noise. The first argument for using the synthetic data as a sanity check is the large amount of training time of the offline and online HWR approaches. Second, as stated by reviewer 5uxE, by evaluating our method on synthetic and real-world datasets, it is more reasonable to argue that the method is applicable to real-world scenarios.
>
> (W5) Indeed, no other paper solves this specific task, but we can compare to other recent methods such as mixing, contrastive losses or proxy-based losses. We will run further experiments with these losses. The large amount of training time required for five cross-validation splits and two datasets makes a fast revision of the paper difficult. Hence, we suggest a resubmission as also suggested by reviewer cbhN.
>
> (W6) We will use common notations and replace the class label y, the image dimension o and p, and the feature space dimension q and w.
>
> (W7) We will rewrite the Abstract and focus more on the online HWR task than defining the task of cross-modal learning.
>
> (W8) Reviewer 6y6a is right that in Figure 4 the latent representation means the output of the convolutional layers, which is misleadingly presented in Figure 4.

---

### Review · Reviewer_5uxE · 2022-06-03

**Summary Of Contributions:**

This paper proposes to use the triplet loss for the integrated representation learning between image and time-series data. The loss uses positive and negative identities to create sample pairs with different labels. Empirical results on synthetic data and handwriting recognition data from sensor-enhanced pens are provided.



**Broader Impact Concerns:**

I do not have particular concerns on the broader impacts.

**Requested Changes:**

The authors may consider revising this paper according to the listed weaknesses.

**Strengths And Weaknesses:**


Strengths:

(1)	The experiments are conducted on both real and synthetic data.


Weaknesses:

(1)	The writing and presentation of this paper require fundamental improvement. I feel truly hard to understand the main idea. In fact, the contributions of this paper are ambiguous in its current form.

(2)	There are too many acronyms.

(3)	Figure 4 is complicated and not clear. The unnecessary details (e.g., network architectures of ConvNets) can be omitted and this main contribution of this paper needs to be highlighted.

(4)	In Figure 7, it seems that a high accuracy of handwriting recognition has been attained by only leveraging the image-based data. Why do we need cross-modal learning?

---

> ### Author Response · Authors · 2022-06-03
> **Answer to feedback from reviewer 5uxE**
>
> We would like to thank the reviewer for the comments. We address the statements as following.
>
> (W1)  As stated in the official comment (1), we will clarify the contributions of this paper and make them more understandable. We will improve the writing and presentation of the paper.
>
> (W2)  We understand that there are many acronyms that make the reading of the paper hard. These acronyms are required to formulate the triplet loss well-arranged. We will replace the  abbreviation of the shared loss to make it more clear. Furthermore, as suggested by reviewer cbhN, we omit the acronym CRL.
>
> (W3)  We will omit unnecessary details of Figure 4 such as the network details to improve the readability of the figure. For reproducibility, we add the details of the network in the appendix.
>
> (W4)  Indeed, the model achieves a higher accuracy on the synthetic image data than for the time-series synthetic data, which is presented in Figure 7. We used the synthetic dataset to show baseline improvements. Cross-modal learning is required when both data sources (image and time-series data) are present. For the application of online handwriting recognition, both data sources can be exploited in a cross-modal setup at training time, while only the time-series data is available at inference time (learning with privileged information). For this application, a cross-modal learning technique is required.

---

### Author Response · Authors · 2022-06-03
**Summarizing statements from all reviewers**

We would like to thank the reviewers for their comments and for taking the time to point out options to improve our manuscript. In the following, we summarize comments that are mentioned by two or more reviewers.

(C1) From all reviewers we got the feedback that the presentation of the paper is not clear.  Particularly, we will revise the writing of the manuscript to make the contributions of our paper clearer.

(C2) Reviewer 6y6a and reviewer cbhN suggest focusing more on the task of online HWR and the challenge of cross-modal learning with time-series data. We will rephrase the paper as such and focus on the peculiarities of the application.

(C3) Reviewer 6y6a and reviewer cbhN mentioned that the paper is lacking important related work. We add the suggested papers for cross-modal learning and the triplet loss and show differences to state-of-the-art work in the related work section. We will make use of further contrastive learning techniques to compare with our triplet learning approach. While we presented related work for offline handwriting recognition (HWR) in the appendix, we will add further related work for online HWR. From these, we will use selected methods for comparison to state-of-the-art HWR.

(C4) As suggested by reviewer 5uxE and reviewer 6y6a, we will reduce the number of acronyms, i.e., we will replace the non-common acronym CRL. Furthermore, we will separate the use of DML and continual learning.

---

### Decision · Action_Editors · 2022-07-02

**Recommendation:** Reject

**Comment:**

The reviewers raised unanimous concerns about improper and insufficient acknowledgement of prior work on cross-modal metric learning using triplet losses. Furthermore, there are numerous concerns about the quality of presentation and writing. In my opinion these concerns require a major overhaul of the paper and another round of full reviews. Although the authors provided the rebuttal, no revised PDF is submitted during the discussion period. This makes it unclear how the concerns will be addressed in the paper. I concur with the reviewers' final recommendations to reject this submission in its current form.